# Computing Approximate $\ell_p$ Sensitivities

**Swati Padmanabhan**
Massachusetts Institute of Technology
pswt@mit.edu

**David P. Woodruff**
Carnegie Mellon University
dwoodruf@cs.cmu.edu

**Qiuyi (Richard) Zhang**
Google Research
qiuyiz@google.com

## Abstract

Recent works in dimensionality reduction for regression tasks have introduced the notion of sensitivity, an estimate of the importance of a specific datapoint in a dataset, offering provable guarantees on the quality of the approximation after removing low-sensitivity datapoints via subsampling. However, fast algorithms for approximating sensitivities, which we show is equivalent to approximate regression, are known for only the $\ell_2$ setting, in which they are popularly termed leverage scores. In this work, we provide the first efficient algorithms for approximating $\ell_p$ sensitivities and related summary statistics of a given matrix. In particular, for a given $n \times d$ matrix, we compute $\alpha$-approximation to its $\ell_1$ sensitivities at the cost of $O(n/\alpha)$ sensitivity computations. For estimating the total $\ell_p$ sensitivity (i.e. the sum of $\ell_p$ sensitivities), we provide an algorithm based on importance sampling of $\ell_p$ Lewis weights, which computes a constant factor approximation to the total sensitivity at the cost of roughly $O(\sqrt{d})$ sensitivity computations. Furthermore, we estimate the maximum $\ell_1$ sensitivity, up to a $\sqrt{d}$ factor, using $O(d)$ sensitivity computations. We generalizeall these results to $\ell_p$ norms for $p > 1$. Lastly, we experimentally show that for a wide class of matrices in real-world datasets, the total sensitivity can be quickly approximated and is significantly smaller than the theoretical prediction, demonstrating that real-world datasets have low intrinsic effective dimensionality.

## 1 Introduction

Many modern large-scale machine learning datasets comprise tall matrices $\mathbf{A} \in \mathbb{R}^{n \times d}$ with $d$ features and $n \gg d$ training examples, often making it computationally expensive to run them through even basic algorithmic primitives. Therefore, many applications spanning dimension reduction, privacy, and fairness aim to estimate the importance of a given datapoint as a first step. Such importance estimates afford us a principled approach to focus our attention on a subset of only the most important examples.

In view of this benefit, several sampling approaches have been extensively developed in the machine learning literature. One of the simplest such methods prevalent in practice is uniform sampling. Prominent examples of this technique include variants of stochastic gradient descent methods [1–4] such as [5–14] developed for the "finite-sum minimization" problem

$$\text{minimize}_{\boldsymbol{\theta} \in \mathcal{X}} \quad \sum_{i=1}^{n} f_i(\boldsymbol{\theta}). \tag{1.1}$$

where, in the context of empirical risk minimization, each $f_i : \mathcal{X} \mapsto \mathbb{R}_{\geq 0}$ in (1.1) measures the loss incurred by the $i$-th data point from the training set, and in generalized linear models, each $f_i$ represents a link function applied to a linear predictor evaluated at the $i$-th data point. The aforementioned algorithms randomly sample a function $f_i$ and make progress via gradient estimation techniques.

37th Conference on Neural Information Processing Systems (NeurIPS 2023).

However, uniform sampling can lead to significant information loss when the number of important training examples is relatively small. Hence, *importance sampling* methods that assign higher sampling probabilities to more important examples have emerged both in theory [15, 16] and practice [17–21]. One such technique is based on *sensitivity scores*, defined [15] for each coordinate $i \in [n]$ as:

$$\boldsymbol{\sigma}_i \stackrel{\text{def}}{=} \max_{\mathbf{x} \in \mathcal{X}} \frac{f_i(\mathbf{x})}{\sum_{j=1}^n f_j(\mathbf{x})}. \tag{1.2}$$

Sampling each $f_i$ independently with probability $p_i \propto \boldsymbol{\sigma}_i$ and scaling the sampled row with $1/p_i$ preserves the objective in (1.1) in expectation for every $\mathbf{x} \in \mathcal{X}$. Further, one gets a $(1 \pm \varepsilon)$-approximation to this objective by sampling $\widetilde{O}(\mathfrak{S}d\varepsilon^{-2})$ functions [22, Theorem 1.1], where $\mathfrak{S} = \sum_{i=1}^n \boldsymbol{\sigma}_i$ is called the *total sensitivity*, and $d$ is an associated VC dimension.

Numerous advantages of sensitivity sampling over $\ell_p$ Lewis weight sampling [23] were highlighted in the recent work [24], prominent among which is that a sampling matrix built using the $\ell_p$ sensitivities of $\mathbf{A}$ is significantly smaller than one built using its $\ell_p$ Lewis weights in many important cases, e.g., when the total sensitivity $\mathfrak{S}$ is small for $p > 2$ (which is seen in many structured matrices, such as combinations of Vandermonde, sparse, and low-rank matrices as studied in [25] as well as those in many naturally occuring datasets (cf. Section 4)). Additionally, sensitivity sampling has found use in constructing $\epsilon$-approximators for shape-fitting functions, also called strong coresets [15, 16, 26–28], clustering [16, 29, 30], logistic regression [31, 32], and least squares regression [33, 34].

Despite their benefits, the computational burden of estimating these sensitivities has been known to be significant, requiring solving an expensive maximization problem for each datapoint, which is a limiting factor in their use [35]. A key exception is for $f_i(\mathbf{x}) = (\mathbf{a}_i^\top \mathbf{x})^2$, when the sensitivity score of $f_i$ is exactly the leverage score of $\mathbf{a}_i$, which can be computed quickly in $O(\log(n))$ sensitivity calculations, as opposed to the naive $O(n)$ sensitivity calculations (cf. Section 2).

## 1.1 Our contributions

In this work, we initiate a systematic study of algorithms for approximately computing the $\ell_p$ sensitivities of a matrix, generally up to a constant factor. We first define $\ell_p$ sensitivities:

**Definition 1.1.** *[15] Given a full-rank matrix $\mathbf{A} \in \mathbb{R}^{n \times d}$ with $n \geq d$ and a scalar $p \in (0, \infty)$, let $\mathbf{a}_i$ denote the $i$'th row of matrix $\mathbf{A}$. Then the vector of $\ell_p$ sensitivities of $\mathbf{A}$ is defined as[1] (cf. Section 2 for the notation)*

$$\boldsymbol{\sigma}_p(\mathbf{a}_i) \stackrel{\text{def}}{=} \max_{\mathbf{x} \in \mathbb{R}^d, \mathbf{A}\mathbf{x} \neq \mathbf{0}} \frac{|\mathbf{a}_i^\top \mathbf{x}|^p}{\|\mathbf{A}\mathbf{x}\|_p^p}, \text{ for all } i \in [n], \tag{1.3}$$

*and the total $\ell_p$ sensitivity is defined as $\mathfrak{S}_p(\mathbf{A}) \stackrel{\text{def}}{=} \sum_{i \in [n]} \boldsymbol{\sigma}_p(\mathbf{a}_i)$.*

These $\ell_p$ sensitivities yield sampling probabilities for $\ell_p$ regression, a canonical optimization problem that captures least squares regression, maximum flow, and linear programming, in addition to appearing in applications like low rank matrix approximation [36], sparse recovery [37], graph-based semi-supervised learning [38–40], and data clustering [41]. However, algorithms for approximating $\ell_p$ sensitivities were known for only $p = 2$ (when they are called "leverage scores"). As is typical in large-scale machine learning, we assume $n \gg d$. We now state our contributions, which are three-fold.

**(1) Approximation algorithms.** We design algorithms for three prototypical computational tasks associated with $\ell_p$ sensitivities of a matrix $\mathbf{A} \in \mathbb{R}^{n \times d}$. Specifically, for an approximation parameter $\alpha > 1$, we can simultaneously estimate all $\ell_p$ sensitivities up to an additive error of $O\left(\frac{\alpha^p}{n} \mathfrak{S}_p(\mathbf{A})\right)$ via $O(n/\alpha)$ individual sensitivity calculations, reducing our runtime by sacrificing some accuracy. We limit most of our results in the main text to $\ell_1$ sensitivities, but extend these to all $p \geq 1$ in Appendix C. To state our results for the $\ell_p$ setting, we use $\mathbf{nnz}(\mathbf{A})$ to denote the number of non-zero entries of the matrix $\mathbf{A}$, $\omega$ for the matrix multiplication constant, and we introduce the following piece of notation.

**Definition 1.2** (Notation for $\ell_p$ Results)**.** *We introduce the notation $\mathsf{LP}(m, d, p)$ to denote the cost of approximating one $\ell_p$ sensitivity of an $m \times d$ matrix up to an accuracy of a given constant factor.*

---

[1]From hereon, for notational conciseness, we omit stating $\mathbf{A}\mathbf{x} \neq \mathbf{0}$ in the problem constraint.

**Theorem 1.3** (**Estimating all sensitivities**; informal version of Theorem 3.1 and Theorem C.2)**.** *Given a matrix* $\mathbf{A} \in \mathbb{R}^{n \times d}$ *and* $1 \leq \alpha \ll n$, *there exists an algorithm (Algorithm 1) which returns a vector* $\widetilde{\boldsymbol{\sigma}} \in \mathbb{R}_{\geq 0}^n$ *such that, with high probability, for each* $i \in [n]$, *we have*

$$\boldsymbol{\sigma}_1(\mathbf{a}_i) \leq \widetilde{\boldsymbol{\sigma}}_i \leq O(\boldsymbol{\sigma}_1(\mathbf{a}_i) + \tfrac{\alpha}{n}\mathfrak{S}_1(\mathbf{A})).$$

*Our algorithm's runtime is* $\widetilde{O}\left(\mathbf{nnz}(\mathbf{A}) + \frac{n}{\alpha} \cdot d^\omega\right)$. *More generally, for* $p \geq 1$, *we can compute (via Algorithm 6) an estimate* $\widetilde{\boldsymbol{\sigma}} \in \mathbb{R}_{\geq 0}^n$ *satisfying* $\boldsymbol{\sigma}_p(\mathbf{a}_i) \leq \widetilde{\boldsymbol{\sigma}}_i \leq O(\alpha^{p-1}\boldsymbol{\sigma}_p(\mathbf{a}_i) + \frac{\alpha^p}{n}\mathfrak{S}_p(\mathbf{A}))$ *with high probability for each* $i \in [n]$, *at a cost of* $O\left(\mathbf{nnz}(\mathbf{A}) + \frac{n}{\alpha} \cdot \mathsf{LP}(O(d^{\max(1,p/2)}), d, p)\right)$.

Two closely related properties arising widely in algorithms research are the *total sensitivity* $\mathfrak{S}_1(\mathbf{A})$ and the *maximum sensitivity* $\|\boldsymbol{\sigma}_1(\mathbf{A})\|_\infty$. As stated earlier, one important feature of the total sensitivity is that it determines the total sample size needed for function approximation, thus making its efficient estimation crucial for fast $\ell_1$ regression. Additionally, it was shown in [24] that the sample complexity of sensitivity sampling is much lower than that of Lewis weight sampling in many practical applications such as when the total sensitivity is small; therefore, an approximation of the total sensitivity can be used as a fast test for whether or not to proceed with sensitivity sampling (involving the costly task of calculating all sensitivities).

While the total sensitivity helps us characterize the whole dataset (as explained above), the *maximum sensitivity* captures the importance of the most important datapoint and is used in, e.g., experiment design and in reweighting matrices for low coherence [42]. Additionally, it captures the maximum extent to which a datapoint can influence the objective function and is therefore used in differential privacy [43]. While one could naively estimate the total and maximum sensitivities by computing all $n$ sensitivities using Theorem 1.3, we give faster algorithms that, strikingly, have no polynomial dependence on $n$. In the assumed regime of $n \gg d$, this is a significant runtime improvement.

**Theorem 1.4** (**Estimating the total sensitivity**; informal version of Theorem 3.5)**.** *Given a matrix* $\mathbf{A} \in \mathbb{R}^{n \times d}$, *a scalar* $p \geq 1$, *and a small constant* $\gamma < 1$, *there exists an algorithm (Algorithm 2), which returns a positive scalar* $\widehat{s}$ *such that, with a probability of at least* $0.99$, *we have*

$$\mathfrak{S}_p(\mathbf{A}) \leq \widehat{s} \leq (1 + O(\gamma))\mathfrak{S}_p(\mathbf{A}).$$

*Our algorithm's runtime is* $\widetilde{O}\left(\mathbf{nnz}(\mathbf{A}) + \frac{1}{\gamma^2} \cdot d^{|p/2-1|}\mathsf{LP}(O(d^{\max(1,p/2)}, d, p)\right)$.

In Algorithm 4 (presented in Appendix B.2), we give an alternate method to estimate $\mathfrak{S}_1(\mathbf{A})$ based on recursive leverage score sampling, without $\ell_p$ Lewis weight calculations, which we believe to be of independent interest. All of our algorithms utilize approximation properties of $\ell_p$ Lewis weights and subspace embeddings and recent developments in fast computation of these quantities.

**Theorem 1.5** (**Estimating the maximum sensitivity**; informal version of Theorem 3.7 and Theorem C.3)**.** *Given a matrix* $\mathbf{A} \in \mathbb{R}^{n \times d}$, *there exists an algorithm (Algorithm 3) which returns a scalar* $\widetilde{\boldsymbol{\sigma}} > 0$ *such that*

$$\|\boldsymbol{\sigma}_1(\mathbf{A})\|_\infty \leq \widetilde{\boldsymbol{\sigma}} \leq O(\sqrt{d}\|\boldsymbol{\sigma}_1(\mathbf{A})\|_\infty).$$

*Our algorithm's runtime is* $\widetilde{O}(\mathbf{nnz}(\mathbf{A}) + d^{\omega+1})$. *Furthermore, via Algorithm 7, this result holds for all* $p \in [1, 2]$ *and for* $p > 2$, *this guarantee costs* $\widetilde{O}\left(\mathbf{nnz}(\mathbf{A}) + d^{p/2} \cdot \mathsf{LP}(O(d^{p/2}), d, p)\right)$.

**(2) Hardness results.** In the other direction, while it is known that $\ell_p$ sensitivity calculation reduces to $\ell_p$ regression, we show the converse. Specifically, we establish hardness of computing $\ell_p$ sensitivities in terms of the cost of solving multiple corresponding $\ell_p$ regression problems, by showing that a $(1 + \varepsilon)$-approximation to $\ell_p$-sensitivities solves $\ell_p$ multi-regression up to an accuracy factor of $1 + \varepsilon$.

**Theorem 1.6** (**Leave-One-Out $\ell_p$ Multiregression Reduces to $\ell_p$ Sensitivities**; informal[2])**.** *Suppose that we are given an sensitivity approximation algorithm* $\mathcal{A}$, *which for any matrix* $\mathbf{A}' \in \mathbb{R}^{n' \times d'}$ *and accuracy parameter* $\varepsilon' \in (0, 1)$, *computes* $(1 \pm \varepsilon')\boldsymbol{\sigma}_p(\mathbf{A}')$ *in time* $\mathcal{T}(n', d', \mathbf{nnz}(\mathbf{A}'), \varepsilon')$. *Given a matrix* $\mathbf{A} \in \mathbb{R}^{n \times d}$ *with* $n \geq d$, *let* $OPT_i := \min_{\mathbf{y} \in \mathbb{R}^{d-1}} \|\mathbf{A}_{:-i}\mathbf{y} + \mathbf{A}_{:i}\|_p^p$ *and* $\mathbf{y}_i^\star := \arg\min_{\mathbf{y} \in \mathbb{R}^{d-1}} \|\mathbf{A}_{:-i}\mathbf{y} + \mathbf{A}_{:i}\|_p$ *for all the* $i \in [d]$. *Then, there exists an algorithm that takes* $\mathbf{A} \in \mathbb{R}^{n \times d}$ *and computes* $(1 \pm \varepsilon)OPT_i$ *for all* $i$ *in time* $\mathcal{T}(n + d, d, \mathbf{nnz}(\mathbf{A}), \varepsilon)$.

---

[2]Please see Lemma D.2 for the formal version with a small additive error.

For $p \neq 2$, approximating all $\ell_p$ sensitivities to a high accuracy would therefore require improving $\ell_p$ multi-regression algorithms with $\Omega(d)$ instances, for which the current best high-accuracy algorithms take $\text{poly}(d) \, \mathbf{nnz}(\mathbf{A})$ time [44, 45]. More concretely, our hardness results imply that in general one cannot compute all sensitivities as quickly as leverage scores unless there is a major breakthrough in multi-response regression. In order to show this, we introduce a reduction algorithm that efficiently regresses columns of $\mathbf{A}$ against linear combinations of other columns of $\mathbf{A}$, by simply adding a row to $\mathbf{A}$ capturing the desired linear combination and then computing sensitivities, which will reveal the cost of the corresponding regression problem. We can solve $\Theta(d)$ of these problems by augmenting the matrix to increase the rows and columns by at most a constant factor. We defer these details to Appendix D.

**(3) Numerical experiments.** We consolidate our theoretical contributions with a demonstration of the empirical advantage of our approximation algorithms over the naive approach of estimating these sensitivities using the UCI Machine Learning Dataset Repository [46] in Section 4. We found that many datasets have extremely small total sensitivities; therefore, fast sensitivity approximation algorithms utilize this small intrinsic dimensionality of real-world data far better than Lewis weights sampling, with sampling bounds often a factor of 2-5x better. We also found these sensitivities to be easy to approximate, with the accuracy-runtime tradeoff much better than our theory suggests, with up to 40x faster in runtime. Lastly, we show that our algorithm for estimating the total sensitivity produces accurate estimates, up to small constant factors, with a runtime speedup of at least 4x.

## 1.2 Related work

Introduced in the pioneering work of [15], sensitivity sampling falls under the broad framework of "importance sampling". It has found use in constructing $\epsilon$-approximators of numerical integrands of several function classes in numerical analysis, statistics (e.g., in the context of VC dimension), and computer science (e.g., in coresets for clustering). The results from [15] were refined and extended by [16], with more advances in the context of shape-fitting problems in [26–28], clustering [16, 29, 30], logistic regression [31, 32], least squares regression [33, 34, 47], principal component analysis [48], reinforcement learning [49, 50], and pruning of deep neural networks [51–53]. Sampling algorithms for $\ell_p$ subspace embeddings have also been extensively studied in functional analysis [54–59] as well as in theoretical computer science [23, 60, 44, 45].

Another notable line of work in importance sampling uses Lewis weights to determine sampling probabilities. As we explain in Section 2, Lewis weights are closely related to sensitivities. Many modern applications of Lewis weights in theoretical computer science we introduced in [23], which gave input sparsity time algorithms for approximating Lewis weights and used them to obtain fast algorithms for solving $\ell_p$ regression. They have subsequently been used in row sampling algorithms for data pre-processing [33, 61, 62, 60, 23], computing dimension-free strong coresets for $k$-median and subspace approximation [63], and fast tensor factorization in the streaming model [64]. Lewis weights are also used for $\ell_1$ regression in: [65] for stochastic gradient descent pre-conditioning, [66] for quantile regression, and [67] to provide algorithms for linear algebraic problems in the sliding window model. Lewis weights have also become widely used in convex geometry [68], randomized numerical linear algebra [23, 42, 69, 64, 70], and machine learning [71–73].

## 2 Notation and preliminaries

We use boldface uppercase and lowercase letters for matrices and vectors respectively. We denote the $i^{\text{th}}$ row vector of a matrix $\mathbf{A}$ by $\mathbf{a}_i$. Given a matrix $\mathbf{M}$, we use $\text{Tr}(\mathbf{M})$ for its trace, $\text{rank}(\mathbf{M})$ for its rank, $\mathbf{M}^\dagger$ for its Moore-Penrose pseudoinverse, and $|\mathbf{M}|$ for the number of its rows. When $x$ is an integer, we use $[x]$ for the set of integers $1, 2, \ldots, x$. For two positive scalars $x$ and $y$ and a scalar $\alpha \in (0, 1)$, we use $x \approx_\alpha y$ to denote $(1 - \alpha)y \leq x \leq (1 + \alpha)y$; we sometimes use $x \approx y$ to mean $(1 - c)y \leq x \leq (1 + c)y$ for some appropriate universal constant $c$. We use $\widetilde{O}$ to hide dependence on $n^{o(1)}$. We acknowledge that there is some notation overloading between $p$ (when used to denote the scalar in, for example, $\ell_p$ norms) and $p_i$ (when used to denote some probabilities) but the distinction is clear from context. We defer the proofs of facts stated in this section to Appendix A.

**Notation for sensitivities.** We start with a slightly general definition of sensitivities: $\boldsymbol{\sigma}_p^{\mathbf{B}}(\mathbf{a}_i)$ to denote the $\ell_p$ sensitivity of $\mathbf{a}_i$ with respect to $\mathbf{B}$; when computing the sensitivity with respect to the

same matrix that the row is drawn from, we omit the matrix superscript:

$$\boldsymbol{\sigma}_p^{\mathbf{B}}(\mathbf{a}_i) := \max_{\mathbf{x}} \frac{|\mathbf{a}_i^\top \mathbf{x}|^p}{\|\mathbf{B}\mathbf{x}\|_p^p} \text{ and } \boldsymbol{\sigma}_p(\mathbf{a}_i) := \max_{\mathbf{x}} \frac{|\mathbf{a}_i^\top \mathbf{x}|^p}{\|\mathbf{A}\mathbf{x}\|_p^p}. \tag{2.1}$$

When referring to the vector of $\ell_p$ sensitivities of all the rows of a matrix, we omit the subscript in the argument: so, $\boldsymbol{\sigma}_p^{\mathbf{B}}(\mathbf{A})$ is the vector of $\ell_p$ sensitivities of rows of the matrix $\mathbf{A}$, each computed with respect to the matrix $\mathbf{B}$ (as in Equation (2.1)), and analogously, $\boldsymbol{\sigma}_p(\mathbf{A})$ is the vector of $\ell_p$ sensitivities of the rows of matrix $\mathbf{A}$, each computed with respect to itself. Similarly, the total sensitivity is the sum of $\ell_p$ sensitivities of the rows of $\mathbf{A}$ with respect to matrix $\mathbf{B}$, for which we use the following notation:

$$\mathfrak{S}_p^{\mathbf{B}}(\mathbf{A}) := \sum_{i \in [|\mathbf{A}|]} \boldsymbol{\sigma}_p^{\mathbf{B}}(\mathbf{a}_i) \text{ and } \mathfrak{S}_p(\mathbf{A}) := \sum_{i \in [|\mathbf{A}|]} \boldsymbol{\sigma}_p(\mathbf{a}_i), \tag{2.2}$$

where, analogous to the rest of the notation regarding sensitivities, we omit the superscript when the sensitivities are being computed with respect to the input matrix itself.

While $\boldsymbol{\sigma}_p^{\mathbf{B}}(\mathbf{a}_i)$ could be infinite, by appending $\mathbf{a}_i$ to $\mathbf{B}$, the generalized sensitivity becomes once again contained in $[0, 1]$. Specifically, a simple rearrangement of the definition gives us the identity $\boldsymbol{\sigma}_p^{\mathbf{B} \cup \mathbf{a}_i}(\mathbf{a}_i) = 1/(1 + 1/\boldsymbol{\sigma}_p^{\mathbf{B}}(\mathbf{a}_i))$. We now define leverage scores, which are a special type of sensitivities and also amenable to efficient computation [60], which makes them an ideal "building block" in algorithms for approximating sensitivities.

**Leverage scores are $\ell_2$ sensitivities.** The leverage score of the $i^{\text{th}}$ row $\mathbf{a}_i$ of $\mathbf{A}$ is $\tau_i(\mathbf{A}) \overset{\text{def}}{=} \mathbf{a}_i^\top (\mathbf{A}^\top \mathbf{A})^\dagger \mathbf{a}_i$. The $i^{\text{th}}$ leverage score is also the $i^{\text{th}}$ diagonal entry of the orthogonal projection matrix $\mathbf{A}(\mathbf{A}^\top \mathbf{A})^\dagger \mathbf{A}^\top$. Since the eigenvalues of an orthogonal projection matrix are zero or one, it implies $\tau_i(\mathbf{A}) = \mathbf{1}_i^\top \mathbf{A}(\mathbf{A}^\top \mathbf{A})^\dagger \mathbf{A}^\top \mathbf{1}_i \le 1$ for all $i \in [n]$ and $\sum_{i=1}^n \tau_i(\mathbf{A}) \le d$ (see [74]).

It turns out that $\tau_i(\mathbf{A}) = \boldsymbol{\sigma}_2(\mathbf{a}_i)$ (see [75]). This may be verified by applying a change of basis to the variable $\mathbf{y} = \mathbf{A}\mathbf{x}$ in the definition of sensitivity and working out that the maximizer is the vector parallel to $\mathbf{y} = \mathbf{A}^\dagger \mathbf{a}_i$. This also gives an explicit formula for the total $\ell_2$ sensitivity: $\mathfrak{S}_2(\mathbf{A}) = \text{rank}(\mathbf{A})$.

The fastest algorithm for constant factor approximation to leverage scores is by [60], as stated next.

**Fact 2.1** ([60, Lemma 7]). *Given a matrix $\mathbf{A} \in \mathbb{R}^{n \times d}$, we can compute constant-factor approximations to all its leverage scores in time $O(\mathbf{nnz}(\mathbf{A}) \log(n) + d^\omega \log(d))$.*

In this paper, the above is the result we use to compute leverage scores (to a constant accuracy). Indeed, since the error accumulates multiplicatively across different steps of our algorithms, a high-accuracy algorithm for leverage score computation does not serve any benefit over this constant-accuracy one. Leverage scores approximate $\ell_1$ sensitivities up to a distortion factor of $\sqrt{n}$, as seen from the below known fact we prove, for completeness, in Appendix A.

**Fact 2.2** (Crude sensitivity approximation via leverage scores). *Given a matrix $\mathbf{A} \in \mathbb{R}^{n \times d}$, let $\boldsymbol{\sigma}_1(\mathbf{a}_i)$ denote the $i^{\text{th}}$ $\ell_1$ sensitivity of $\mathbf{A}$ with respect to $\mathbf{A}$, and let $\tau_i(\mathbf{A})$ denote its $i^{\text{th}}$ leverage score with respect to $\mathbf{A}$. Then we have $\sqrt{\frac{\tau_i(\mathbf{A})}{n}} \le \boldsymbol{\sigma}_1(\mathbf{a}_i) \le \sqrt{\tau_i(\mathbf{A})}$.*

In general, $\ell_p$ sensitivities of rows of a matrix suffer a distortion factor of at most $n^{|1/2-1/p|}$ from corresponding leverage scores, which follows from Hölder's inequality. The $\ell_p$ generalizations of leverages scores are called $\ell_p$ Lewis weights [54, 23], which satisfy the recurrence $\mathbf{w}_i = \tau_i(\mathbf{W}^{1/2-1/p}\mathbf{A})$. However, unlike when $p = 2$, $\ell_p$ Lewis weights are *not* equal to sensitivities, and from an approximation perspective, they provide only a one-sided bound on the sensitivities $\boldsymbol{\sigma}_p^{\mathbf{A}}(\mathbf{a}_i) \le d^{\max(0, p/2-1)} \mathbf{w}_p^{\mathbf{A}}(\mathbf{a}_i)$ [75, 76]. While we have algorithms for efficiently computing leverage scores [60] and Lewis weights [23, 77], extensions for utilizing these ideas for fast and accurate sensitivity approximation algorithms for all $p$ have been limited, which is the gap this paper aims to fill. A key regime of interest for many downstream algorithms is a small constant factor approximation to true sensitivities, such as for subsampling [78].

**Subspace embeddings.** We use sampling-based constant-approximation $\ell_p$ subspace embeddings that we denote by $\mathbf{S}_p$. When the type of embedding is clear from context, we omit the subscript.

**Definition 2.3** ($\ell_p$ **Subspace Embedding**). *Given* $\mathbf{A} \in \mathbb{R}^{n \times d}$ *(typically $n \gg d$), we call $\mathbf{S} \in \mathbb{R}^{r \times n}$ (typically $d \leq r \ll n$) an $\epsilon$-approximate $\ell_p$ subspace embedding of $\mathbf{A}$ if $\|\mathbf{SAx}\|_p \approx_\varepsilon \|\mathbf{Ax}\|_p \ \forall \mathbf{x} \in \mathbb{R}^d$.*

When the diagonal matrix $\mathbf{S}$ is defined as $\mathbf{S}_{ii} = 1/\sqrt{p_i}$ with probability $p_i = \Omega(\tau_i(\mathbf{A}))$ (and 0 otherwise), then with high probability $\mathbf{SA}$ is an $\ell_2$ subspace embedding for $\mathbf{A}$ [33, 79]; further, $\mathbf{S}$ has at most $\widetilde{O}(d\varepsilon^{-2})$ non-zero entries. For general $p$, we use Lewis weights to construct $\ell_p$ subspace embeddings fast [23, Theorem 7.1]. This fast construction has been made possible by the reduction of Lewis weight computation to a few (polynomial in $p$) leverage score computations for $p < 4$ [23] as well as $p \geq 4$ [77]. This efficient construction is the reason for our choice of this subspace embedding. We note that there has been an extensive amount of literature on designing $\ell_1$ subspace embeddings for example using Cauchy [80] and exponential [81] random variables.

For $p \leq 2$, it is known that the expected sample complexity of rows is $\sum_{i=1}^n p_i = O(d)$ [34], and for $p > 2$, this is known to be at most $O(d^{p/2})$ [35]. While these bounds are optimal in the worst case, one important application of faster sensitivity calculations is to compute the minimal subspace embedding dimension for average case applications, particularly those seen in practice. This gap is captured by the total sensitivity $\mathfrak{S}_p(\mathbf{A})$, which has been used to perform more sample efficient subspace embeddings [67, 27, 28], as sampling $\widetilde{O}(\mathfrak{S}_p(\mathbf{A})d/\epsilon^2)$ rows suffice to provide an $\ell_p$ subspace embedding.

**Accurate sensitivity calculations.** While leverage scores give a crude approximation, constant-factor approximations of $\ell_p$ sensitivities seem daunting since we must compute worst-case ratios over the input space. However, we can compute the cost of one sensitivity specific row by reducing it to $\ell_p$ regression since by scale invariance, we have $\frac{1}{\sigma_p^{\mathbf{B}}(\mathbf{a}_i)} = \min_{\mathbf{a}_i^\top \mathbf{x} = 1} \|\mathbf{Bx}\|_p^p$.

This reduction allows us to use recent developments in fast approximate algorithms for $\ell_p$ regression [44, 25, 45], which utilize subspace embeddings to first reduce the number of rows of the matrix to $O(\text{poly}(d))$. For the specific case of $p = 1$, which is the focus of the main body of our paper, we may reduce to approximate $\ell_1$ regression on a $O(d) \times d$ matrix, which is a form of linear programming.

**Fact 2.4.** *Given an $n \times d$ matrix, the cost of computing $k$ of its $\ell_1$ sensitivities is $\widetilde{O}(\mathbf{nnz}(\mathbf{A}) + k \cdot d^\omega)$.*

## 3 Approximating functions of sensitivities

We first present a constant-probability algorithm approximating the $\ell_1$ sensitivities in Algorithm 1. Our algorithm is a natural one: Since computing the $\ell_1$ sensitivities of all the rows simultaneously is computationally expensive, we instead hash $\alpha$-sized blocks of random rows into smaller (constant-sized) buckets and compute the sensitivities of the smaller, $O(n/\alpha)$-sized matrix $\mathbf{P}$ so generated, computing the sensitivities with respect to $\mathbf{SA}$, the $\ell_1$ subspace embedding of $\mathbf{A}$. Running this algorithm multiple times gives our high-probability guarantee via the standard median trick.

**Theorem 3.1.** *Given a full-rank matrix $\mathbf{A} \in \mathbb{R}^{n \times d}$ and an approximation factor $1 < \alpha \ll n$, let $\boldsymbol{\sigma}_1(\mathbf{a}_i)$ be the $\ell_1$ sensitivity of the $i^{\text{th}}$ row of $\mathbf{A}$. Then there exists an algorithm that, in time $\widetilde{O}\left(\mathbf{nnz}(\mathbf{A}) + \frac{n}{\alpha} \cdot d^\omega\right)$, returns $\widetilde{\boldsymbol{\sigma}} \in \mathbb{R}^n_{\geq 0}$ such that with high probability, for each $i \in [n]$,*

$$\boldsymbol{\sigma}_1(\mathbf{a}_i) \leq \widetilde{\boldsymbol{\sigma}}_i \leq O(\boldsymbol{\sigma}_1(\mathbf{a}_i) + \tfrac{\alpha}{n}\mathfrak{S}_1(\mathbf{A})). \tag{3.1}$$

*Proof Sketch of Theorem 3.1; full proof in Appendix B.1.* We achieve our guarantee via Algorithm 1. To see this, first note that $\|\mathbf{SAx}\|_1 \approx \Theta(\|\mathbf{Ax}\|_1)$ (since as per Line 1, $\mathbf{SA}$ is an $\ell_1$ subspace embedding of $\mathbf{A}$). Combining this with the fact that every row in $\mathbf{A}$ is mapped, via the designed randomness, to some row in the matrix $\mathbf{P}$ helps establish the desired approximation guarantee. The runtime follows from using Fact 2.4 to compute $\ell_1$ sensitivities with respect to $\mathbf{SA}$ of $|\mathbf{P}| = O(n/\alpha)$ rows. $\square$

As seen in Appendix C.1, our techniques described above also generalize to the $p \geq 1$ case. Specifically, in Theorem C.2, we show that reducing $\ell_p$ sensitivity calculations by an $\alpha$ factor gives an approximation guarantee of the form $O(\alpha^{p-1}\boldsymbol{\sigma}_p(\mathbf{a}_i)) + \frac{\alpha^p}{n}\mathfrak{S}_p(\mathbf{A})$.

**Remark 3.2.** *The sensitivities returned by our algorithm are approximate, with relative and additive error, but are useful in certain settings as they preserve $\ell_p$ regression approximation guarantees while increasing total sample complexity by only a small $\text{poly}(d)$ factor compared to true sensitivities while still keeping it much smaller than that obtained via Lewis weights. To see this with a simple toy*

---

**Algorithm 1** Approximating $\ell_1$-Sensitivities: Row-wise Approximation

---

**Input:** Matrix $\mathbf{A} \in \mathbb{R}^{n \times d}$, approximation factor $\alpha \in (1, n)$
**Output:** Vector $\widetilde{\boldsymbol{\sigma}} \in \mathbb{R}^n_{>0}$ that satisfies, for each $i \in [n]$, with probability 0.9, that

$$\boldsymbol{\sigma}_1(\mathbf{a}_i) \leq \widetilde{\boldsymbol{\sigma}}_i \leq O(\boldsymbol{\sigma}_1(\mathbf{a}_i) + \mathfrak{S}_1(\mathbf{A})\tfrac{\alpha}{n})$$

1: Compute for $\mathbf{A}$ an $\ell_1$ subspace embedding $\mathbf{SA} \in O(d) \times d$ (cf. Definition 2.3)
2: Partition $\mathbf{A}$ into $\frac{n}{\alpha}$ blocks $\mathbf{B}_1, \ldots, \mathbf{B}_{n/\alpha}$ each comprising $\alpha$ randomly selected rows
3: **for** the $\ell^{\text{th}}$ block $\mathbf{B}_\ell$, with $\ell \in [\frac{n}{\alpha}]$ **do**
4:      Sample $\alpha$-dimensional independent Rademacher vectors $\mathbf{r}_1^{(\ell)}, \ldots, \mathbf{r}_{100}^{(\ell)}$
5:      For each $j \in [100]$, compute the row vectors $\mathbf{r}_j^{(\ell)} \mathbf{B}_\ell \in \mathbb{R}^d$
6: **end for**
7: Let $\mathbf{P} \in \mathbb{R}^{100\frac{n}{\alpha} \times d}$ be the matrix of all vectors from Line 5. Compute $\boldsymbol{\sigma}_1^{\mathbf{SA}}(\mathbf{P})$ using Fact 2.4
8: **for** each $i \in [n]$ **do**
9:      Denote by $J$ the set of row indices in $\mathbf{P}$ that $\mathbf{a}_i$ is mapped to in Line 5
10:      Set $\widetilde{\boldsymbol{\sigma}}_i = \max_{j \in J}(\boldsymbol{\sigma}_1^{\mathbf{SA}}(\mathbf{p}_j))$
11: **end for**
12: **Return** $\widetilde{\boldsymbol{\sigma}}$

---

*example, consider the case $p > 2$. Note that by Theorem 1.5 of [24], the sample complexity using approximate sensitivities is $\widetilde{O}(\alpha^{2p-2}\mathfrak{S}_p^{2-2/p}(\mathbf{A}))$, using the $\alpha^p$ factor blowup via Theorem C.2, that using true sensitivities is $\widetilde{O}(\mathfrak{S}_p^{2-2/p}(\mathbf{A}))$, and that using $\ell_p$ Lewis weights is $O(d^{p/2})$. Suppose also that the total sensitivity $\mathfrak{S}_p(\mathbf{A}) = d$. Further assume that $n = d^{10}$ and $\alpha = n^{\frac{1}{10p}} = d^{1/p}$; then the dimension-dependence of sample complexities given by our approximate sensitivities, true sensitivities, and Lewis weights are, respectively, $\widetilde{O}(d^4), \widetilde{O}(d^2)$, and $\widetilde{O}(d^{p/2})$ (ignoring small factors in the exponent that do not affect the asymptotic analysis). Thus, our approximate sensitivities provide a tradeoff in computational cost and sample complexity between using* true *sensitivities and Lewis weights.*

### 3.1 Estimating the sum of $\ell_p$ sensitivities

In this section, we present a one-shot algorithm that provides a constant-approximation to the total $\ell_p$ sensitivity for all $p \geq 1$. This algorithm is based on importance sampling using $\ell_p$ Lewis weights. For our desired guarantees, we crucially need the approximation bounds of $\ell_p$ Lewis weights, as provided by Fact 3.3, efficient computation of $\ell_p$ Lewis weights as provided by [23] for $p < 4$ and by [77] for $p \geq 4$, and the fact that a random $\ell_p$ sampling matrix $\mathbf{S}$ (cf. Fact 3.4) yields $\mathbf{SA}$, a constant-approximation subspace embedding to $\mathbf{A}$ [24, Theorem 1.8].

---

**Algorithm 2** Approximating the Sum of $\ell_p$-Sensitivities for $p \geq 1$

---

**Input:** Matrix $\mathbf{A} \in \mathbb{R}^{n \times d}$, approximation factor $\gamma \in (0.01, 1)$, and a scalar $p \geq 1$
**Output:** A positive scalar that satisfies, with probability 0.99, that

$$\mathfrak{S}_p(\mathbf{A}) \leq \widehat{s} \leq (1 + O(\gamma))\mathfrak{S}_p(\mathbf{A})$$

1: Compute all $\ell_p$ Lewis weights of $\mathbf{A}$, denoting the $i^{\text{th}}$ weight by $\mathbf{w}_p(\mathbf{a}_i)$
2: Define the sampling vector $v \in \mathbb{R}^n_{\geq 0}$ by $v_i = \frac{\mathbf{w}_p(\mathbf{a}_i)}{d}$ for all $i \in [n]$
3: Sample $m = O(d^{|1-p/2|})$ rows with replacement, picking the $i^{\text{th}}$ row with a probability of $v_i$
4: Construct a random $\ell_p$ sampling matrix $\mathbf{S}_p\mathbf{A}$ with probabilities $\{v_i\}_{i=1}^n$ (cf. Fact 3.4)
5: For each sampled row $i_j$, $j \in [m]$, compute $r_j = \frac{\boldsymbol{\sigma}_p^{\mathbf{S}_p\mathbf{A}}(\mathbf{a}_{i_j})}{v_{i_j}}$
6: Return $\frac{1}{m}\sum_{j=1}^m r_j$

---

**Fact 3.3** (Lewis Weights Bounds Sensitivities, [42, 82]). *Given a matrix $\mathbf{A} \in \mathbb{R}^{n \times d}$ and $p \geq 1$, the $\ell_p$ sensitivity $\boldsymbol{\sigma}_p^{\mathbf{A}}(\mathbf{a}_i)$ and $\ell_p$ Lewis weight $\mathbf{w}_p(\mathbf{a}_i)$ of the row $\mathbf{a}_i$ satisfy, for all $i \in [n]$,*

$$\boldsymbol{\sigma}_p^{\mathbf{A}}(\mathbf{a}_i) \leq d^{\max(0, p/2-1)}\, \mathbf{w}_p(\mathbf{a}_i).$$

**Fact 3.4** (Sampling via Lewis Weights [23, 35]). *Given $\mathbf{A} \in \mathbb{R}^{n \times d}$ and $p > 0$. Consider a random diagonal $\ell_p$ sampling matrix $\mathbf{S} \in \mathbb{R}^{n \times n}$ with sampling probabilities $\{p_i\}$ proportional to the $\ell_p$ Lewis weight of $\mathbf{A}$, i.e., for each $i \in [n]$, the $i^{\text{th}}$ diagonal entry is independently set to be*

$$\mathbf{S}_{i,i} = \begin{cases} 1/p_i^{1/p} & \text{with probability } p_i \\ 0 & \text{otherwise} \end{cases}.$$

*Then, with high probability, $\mathbf{S}$ with $O(\varepsilon^{-2} d^{\max(1,p/2)} (\log d)^2 \log(d/\varepsilon))$ rows is an $\ell_p$ subspace embedding for $\mathbf{A}$ (cf. Definition 2.3).*

**Theorem 3.5.** *Given a matrix $\mathbf{A} \in \mathbb{R}^{n \times d}$ and an approximation factor $\gamma \in (0, 1)$, there exists an algorithm, which returns a positive scalar $\widehat{s}$ such that, with a probability $0.99$, we have*

$$\mathfrak{S}_p(\mathbf{A}) \le \widehat{s} \le (1 + O(\gamma))\mathfrak{S}_p(\mathbf{A}).$$

*Our algorithm's runtime is $\widetilde{O}\left(\mathbf{nnz}(\mathbf{A}) + \frac{1}{\gamma^2} \cdot d^{|p/2-1|} \mathsf{LP}(O(d^{\max(1,p/2)}, d, p)\right)$.*

*Proof.* Without loss of generality, we may assume $\mathbf{A}$ to be full rank. Then, its Lewis weights satisfy $\sum_{i=1}^n \mathbf{w}_p(\mathbf{a}_i) = d$. Per Line 2 of Algorithm 2, our sampling distribution is chosen to be $v_i = \frac{\mathbf{w}_p(\mathbf{a}_i)}{d}$ for all $i \in [n]$. We sample from $\mathbf{A}$ rows with replacement, with row $i$ picked with a probability of $v_i$. From the definition of $v_i$ in Line 2 and $r_j$ in Line 5 and the fact that $\mathbf{S}_p\mathbf{A}$ is a constant factor $\ell_p$ subspace embedding of $\mathbf{A}$, our algorithm's output satisfies the following unbiasedness condition:

$$\mathbb{E}\left(\frac{1}{m}\sum_{j\in[m]} r_j\right) = \mathbb{E}\left(\frac{1}{m}\sum_{j=1}^m\sum_{i_j=1}^n \frac{\boldsymbol{\sigma}_p^{\mathbf{S}_p\mathbf{A}}(\mathbf{a}_{i_j})}{v_{i_j}} \cdot v_{i_j}\right) = \mathfrak{S}_p^{\mathbf{S}_p\mathbf{A}}(\mathbf{A}) \le 2\mathfrak{S}_p(\mathbf{A}).$$

By independence, we also have the following bound on the variance of $\frac{1}{m}\sum_{j\in[m]} r_j$:

$$\mathrm{Var}\left(\frac{1}{m}\sum_{j\in[m]} r_j\right) \le \frac{1}{m}\sum_{i=1}^n \frac{\boldsymbol{\sigma}_p^{\mathbf{S}_p\mathbf{A}}(\mathbf{a}_i)^2}{v_i} = \frac{d}{m}\cdot\sum_{i=1}^n \frac{\boldsymbol{\sigma}_p^{\mathbf{S}_p\mathbf{A}}(\mathbf{a}_i)^2}{\mathbf{w}_p(\mathbf{a}_i)},$$

with the final step holding by the choice of $v_i$ in Line 2. In the case that $p \ge 2$, we have

$$\mathrm{Var}\left(\frac{1}{m}\sum_{j\in[m]} r_j\right) \le \frac{d}{m}\cdot\sum_{i=1}^n \frac{\boldsymbol{\sigma}_p^{\mathbf{S}_p\mathbf{A}}(\mathbf{a}_i)^2}{\mathbf{w}_p(\mathbf{a}_i)} \le \frac{d}{m}\cdot\sum_{i\in[n]} \boldsymbol{\sigma}_p^{\mathbf{S}_p\mathbf{A}}(\mathbf{a}_i)\cdot d^{p/2-1} \le \frac{2d^{p/2}}{m}\mathfrak{S}_p(\mathbf{A}),$$

where the first inequality uses Fact 3.3. Therefore, applying Chebyshev's inequality on $\frac{1}{m}\sum_{j\in[m]} r_j$ (as defined in Line 5) with $m = O\left(\frac{d^{p/2}}{\mathfrak{S}_p(\mathbf{A})\gamma^2}\right)$ gives us a $\gamma$-multiplicative accuracy in approximating $\mathfrak{S}_p(\mathbf{A})$ (with the desired constant probability). For $p \ge 2$, we additionally have [24, Theorem 1.7] the lower bound $\mathfrak{S}_p(\mathbf{A}) \ge d$, which when plugged into the value of $m$ gives the claimed sample complexity. A similar argument may be applied for the case $p < 2$; specifically, we have that

$$\mathrm{Var}\left(\frac{1}{m}\sum_{j\in[m]} r_j\right) \le \frac{d}{m}\cdot\sum_{i=1}^n \frac{\boldsymbol{\sigma}_p^{\mathbf{S}_p\mathbf{A}}(\mathbf{a}_i)^2}{\mathbf{w}_p(\mathbf{a}_i)} \le \frac{d}{m}\cdot\sum_{i\in[n]} \boldsymbol{\sigma}_p^{\mathbf{S}_p\mathbf{A}}(\mathbf{a}_i) \le \frac{2d}{m}\mathfrak{S}_p(\mathbf{A}),$$

where the second step is by $\mathbf{w}_p(\mathbf{a}_i) \ge \boldsymbol{\sigma}_p(\mathbf{a}_i)$ from Fact 3.3. For $p < 2$, we also have $\mathfrak{S}_p(\mathbf{A}) \ge d^{p/2}$ from [24, Theorem 1.7]. Applying Chebyshev's inequality with this fact gives a sample complexity of $m = O\left(d^{1-p/2}/\gamma^2\right)$. This completes the correctness guarantee.

**Runtime.** We first compute all $\ell_p$ Lewis weights of $\mathbf{A} \in \mathbb{R}^{n \times d}$ up to a constant multiplicative accuracy, the cost of which is $O\left(\frac{1}{1-|1-p/2|}\log(\log(n))\right)$ leverage score computations for $p < 4$ [23] and $O(p^3 \log(np))$ leverage score computations for $p \ge 4$ [77]. Next, we compute $m = O(d^{|1-p/2|})$ sensitivities with respect to $\mathbf{S}_p\mathbf{A}$. From [23, 35], $\mathbf{S}_p$ has, with high probability, $O(d^{p/2}\log d)$ rows when $p > 2$ and $O(d\log d)$ rows when $p \le 2$. Summing the cost of computing these $m$ sensitivities and that of computing the Lewis weights gives the claimed runtime. $\qquad\square$

**Remark 3.6.** *We present in Appendix B.2 an alternate algorithm, Algorithm 4, for estimating the total $\ell_p$ sensitivity for $p = 1$. Algorithm 4 uses recursive computations of leverage scores, in contrast to Algorithm 2 which uses Lewis weights in a one-shot manner, and may be of independent interest.*

## 3.2 Approximating the maximum $\ell_1$ sensitivity

In this section, we present an algorithm that approximates $\|\boldsymbol{\sigma}_1(\mathbf{A})\|_\infty = \max_{i \in [|\mathbf{A}|]} \boldsymbol{\sigma}_1(\mathbf{a}_i)$, the maximum of the $\ell_1$ sensitivities of the rows of a matrix $\mathbf{A}$. As alluded to in Section 1, a first approach to this problem would be to simply estimate all $n$ sensitivities and compute their maximum. To do better than this, one idea, inspired by the random hashing approach of Algorithm 1, is as follows.

If the matrix has a large number of high-sensitivity rows, then, intuitively, the appropriately scaled maximum sensitivity of a uniformly sampled subset of these rows should approximate $\|\boldsymbol{\sigma}_1(\mathbf{A})\|_\infty$. Specifically, assume that the matrix has at least $\alpha$ rows of sensitivity at least $\|\boldsymbol{\sigma}_1(\mathbf{A})\|_\infty/\sqrt{\alpha}$; uniformly sample $n/\alpha$ rows and return $\widetilde{\boldsymbol{\sigma}}_a$, the (appropriately scaled) maximum of $\boldsymbol{\sigma}_1^{\mathbf{S}_1\mathbf{A}}(\mathbf{a}_i)$, for the sampled rows $i$. Then, $\|\boldsymbol{\sigma}_1(\mathbf{A})\|_\infty \leq \widetilde{\boldsymbol{\sigma}}_a \leq O(\sqrt{\alpha}\|\boldsymbol{\sigma}_1(\mathbf{A})\|_\infty)$ with a constant probability. Here the upper bound guarantee is without any condition on the number of rows with large $\ell_1$ sensitivities.

In the other case, if the matrix does *not* have too many high-sensitivity rows, we could estimate the maximum sensitivity by hashing rows into small buckets via Rademacher combinations of uniformly sampled blocks of $\alpha$ rows each (just like in Algorithm 1). Then, $\widetilde{\boldsymbol{\sigma}}_b$, the scaled maximum of the $\ell_1$ sensitivities of these rows satisfies $\|\boldsymbol{\sigma}_1(\mathbf{A})\|_\infty \leq \widetilde{\boldsymbol{\sigma}}_b \leq O(\sqrt{\alpha}\|\boldsymbol{\sigma}_1(\mathbf{A})\|_\infty)$. Here, it is the *lower* bound that comes for free (i.e., without any condition on the number of rows with large $\ell_1$ sensitivities).

Despite the above strategies working for each case, there is no way to combine these approaches without knowledge of whether the input matrix has a enough "high-sensitivity" rows or not. We therefore avoid this approach and instead present Algorithm 3, where we make use of $\mathbf{S}_\infty\mathbf{A}$, an $\ell_\infty$ subspace embedding of $\mathbf{A}$. Our algorithm hinges on the recent development [75, Theorem 1.3] on efficient construction of such an embedding with simply a subset of $\widetilde{O}(d)$ rows of $\mathbf{A}$.

---

**Algorithm 3** Approximating the Maximum of $\ell_1$-Sensitivities

    **Input:** Matrix $\mathbf{A} \in \mathbb{R}^{n \times d}$
    **Output:** Scalar $\widehat{s} \in \mathbb{R}_{\geq 0}$ that satisfies $\|\boldsymbol{\sigma}_1(\mathbf{A})\|_\infty \leq \widehat{s} \leq C \cdot \sqrt{d} \cdot \|\boldsymbol{\sigma}_1(\mathbf{A})\|_\infty$
1: Compute, for $\mathbf{A}$, an $\ell_\infty$ subspace embedding $\mathbf{S}_\infty\mathbf{A} \in \mathbb{R}^{O(d\log^2(d)) \times d}$ such that $\mathbf{S}_\infty\mathbf{A}$ is a subset of the rows of $\mathbf{A}$ [75, Theorem 1.3]
2: Compute, for $\mathbf{A}$, an $\ell_1$ subspace embedding $\mathbf{S}_1\mathbf{A} \in \mathbb{R}^{O(d) \times d}$
3: Return $\sqrt{d}\|\boldsymbol{\sigma}_1^{\mathbf{S}_1\mathbf{A}}(\mathbf{S}_\infty\mathbf{A})\|_\infty$

---

**Theorem 3.7 (Approximating the Maximum of $\ell_1$ Sensitivities).** *Given a matrix $\mathbf{A} \in \mathbb{R}^{n \times d}$, there exists an algorithm, which in time $\widetilde{O}(\mathbf{nnz}(\mathbf{A}) + d^{\omega+1})$, outputs a positive scalar $\widehat{s}$ that satisfies*

$$\Omega(\|\boldsymbol{\sigma}_1(\mathbf{A})\|_\infty) \leq \widehat{s} \leq O(\sqrt{d}\|\boldsymbol{\sigma}_1(\mathbf{A})\|_\infty).$$

*Proof Sketch of Theorem 3.7; full proof in Appendix B.3.* We achieve our guarantee via Algorithm 3. Define $\mathbf{x}^\star$ and $\mathbf{a}_{i^\star}$ as: $\mathbf{x}^\star, i^\star = \arg\max_{\mathbf{x} \in \mathbb{R}^d, i \in [|\mathbf{A}|]} \frac{|\mathbf{a}_i^\top \mathbf{x}|}{\|\mathbf{A}\mathbf{x}\|_1}$. Since $\mathbf{S}_1\mathbf{A}$ is an $\ell_1$ subspace embedding of $\mathbf{A}$, we have, for any $\mathbf{x} \in \mathbb{R}^d$, that $\|\mathbf{S}_1\mathbf{A}\mathbf{x}\|_1 = \Theta(\|\mathbf{A}\mathbf{x}\|_1)$. If $\mathbf{S}_\infty\mathbf{A}$ contains the row $\mathbf{a}_{i^\star}$, then $\|\mathbf{S}_1\mathbf{A}\mathbf{x}\|_1 = \Theta(\|\mathbf{A}\mathbf{x}\|_1)$ implies $\|\boldsymbol{\sigma}_1^{\mathbf{S}_1\mathbf{A}}(\mathbf{S}_\infty\mathbf{A})\|_\infty = \Theta(1)\|\boldsymbol{\sigma}_1(\mathbf{A})\|_\infty$. In the other case, suppose $\mathbf{a}_{i^\star}$ is not included in $\mathbf{S}_\infty\mathbf{A}$. Then we observe that

$$\|\boldsymbol{\sigma}_1^{\mathbf{S}_1\mathbf{A}}(\mathbf{S}_\infty\mathbf{A})\|_\infty = \max_{\mathbf{x} \in \mathbb{R}^d, \mathbf{c}_j \in \mathbf{S}_\infty\mathbf{A}} \frac{\mathbf{c}_j^\top \mathbf{x}}{\|\mathbf{S}_1\mathbf{A}\mathbf{x}\|_1} \geq \max_{\mathbf{x} \in \mathbb{R}^d} \frac{\|\mathbf{S}_\infty\mathbf{A}\mathbf{x}\|_\infty}{\|\mathbf{S}_1\mathbf{A}\mathbf{x}\|_1} = \Theta(1) \max_{\mathbf{x} \in \mathbb{R}^d} \frac{\|\mathbf{S}_\infty\mathbf{A}\mathbf{x}\|_\infty}{\|\mathbf{A}\mathbf{x}\|_1},$$
$$(3.2)$$

where the the second step is by choosing a specific vector in the numerator and the third step uses $\|\mathbf{S}_1\mathbf{A}\mathbf{x}\|_1 = \Theta(\|\mathbf{A}\mathbf{x}\|_1)$. We further have,

$$\max_{\mathbf{x} \in \mathbb{R}^d} \frac{\|\mathbf{S}_\infty\mathbf{A}\mathbf{x}\|_\infty}{\|\mathbf{A}\mathbf{x}\|_1} \geq \frac{\|\mathbf{S}_\infty\mathbf{A}\mathbf{x}^\star\|_\infty}{\|\mathbf{A}\mathbf{x}^\star\|_1} \geq \frac{\|\mathbf{A}\mathbf{x}^\star\|_\infty}{\sqrt{d}\|\mathbf{A}\mathbf{x}^\star\|_1} \geq \frac{|\mathbf{a}_{i^\star}^\top \mathbf{x}^\star|}{\sqrt{d}\|\mathbf{A}\mathbf{x}^\star\|_1} = \frac{1}{\sqrt{d}}\|\boldsymbol{\sigma}_1(\mathbf{A})\|_\infty,, \quad (3.3)$$

where the first step is by choosing $\mathbf{x} = \mathbf{x}^\star$, the second step is by the guarantee of $\ell_\infty$ subspace embedding, and the final step is by definition of $\boldsymbol{\sigma}_1(\mathbf{a}_{i^\star})$. Combining Inequality (3.2) and Inequality (3.3) gives the claimed lower bound on $\|\boldsymbol{\sigma}_1^{\mathbf{S}_1\mathbf{A}}(\mathbf{S}_\infty\mathbf{A})\|_\infty$. The runtime follows from the cost of computing $\mathbf{S}_\infty\mathbf{A}$ from [75] and that of computing $\widetilde{O}(d)$ of $\ell_1$ sensitivities with respect to $\mathbf{S}_1\mathbf{A}$. $\quad\square$

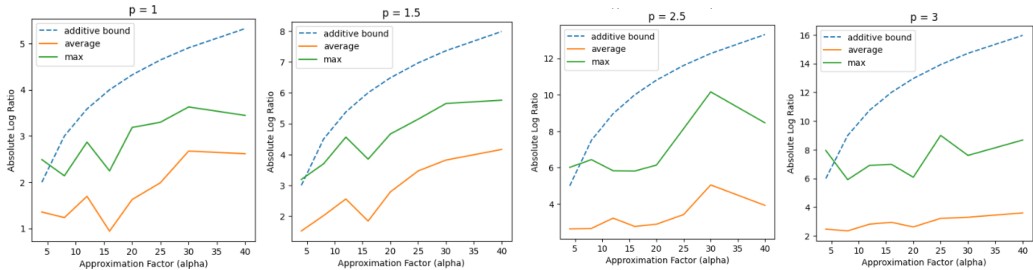

Figure 1: Average absolute log ratios for all $\ell_p$ sensitivity approximations for `wine`, with the theoretical additive bound (dashed).

## 4 Numerical experiments

We demonstrate our fast sensitivity approximations on multiple real-world datasets in the UCI Machine Learning Dataset Repository [46], such as the `wine` and `fires` datasets, for different $p$ and varying approximation parameters $\alpha$. We focus on the `wine` dataset here, with full details in Appendix E.

**Experimental Setup.** For each dataset and each $p$, we first apply the Lewis weight subspace embedding to compute a smaller matrix $\mathbf{SA}$. Then, we run our sensitivity computation Algorithm 1 and measure the average and maximum absolute log ratio $|\log(\sigma_{\text{approx}}/\sigma_{\text{true}})|$ to capture the relative error of the sensitivity approximation. Note that this is stricter than our guarantees, which give only an $O(\alpha)$ additive error; therefore we have no real upper bound on the maximum absolute log ratio due to small sensitivities. We plot an upper bound of $\log(\alpha^p)$, which is the worst case additive error approximation although it does provide relative error guarantees with respect to the average sensitivity. We compare our fast algorithm for computing the total sensitivity with the naive brute-force method by approximating all sensitivities and then averaging.

**Analysis.** In practice, we found most real-world datasets have easy-to-approximate sensitivities with total $\ell_p$ sensitivity about 2-5 times lower than the theoretical upper bound. Figure 1 shows that the average absolute log ratios for approximating the $\ell_p$ sensitivities are much smaller than those suggested by the theoretical upper bound, even for large $\alpha$. Specifically, when $\alpha = 40$, we incur only a 16x accuracy deterioration in exchange for a 40x faster algorithm. This is significantly better than the worst-case accuracy guarantee which for $p = 3$ would be $\alpha^p = 40^3$.

More importantly, we find that empirical total sensitivities are much lower than the theoretical upper bounds suggest, often by a factor of at least 5, especially for $p > 2$. This implies that real-world structure can be utilized for improved data compression and justifies the importance of sensitivity estimation for real-world datasets. Furthermore, our novel algorithm approximates the total sensitivity up to an overestimate within a factor of 1.3, in less than a quarter of the time of the brute-force algorithm. Our observation generalizes to other real-world datasets (see Appendix E).

| $p$ | Total Sensitivity Upper Bound | Brute-Force/True | Approximation | Brute-Force Runtime (s) | Approximate Runtime (s) |
|---|---|---|---|---|---|
| 1 | 14 | 5.2 | 6.4 | 667 | 105 |
| 1.5 | 14 | 11.6 | 14.2 | 673 | 131 |
| 2.5 | 27.1 | 13.8 | 14.9 | 693 | 209 |
| 3 | 52.4 | 7.2 | 8.9 | 686 | 192 |

Table 1: Runtime comparison for computing total sensitivities for the `wine` dataset, which has matrix shape $(177, 14)$.

## Acknowledgements

We are very grateful to: Sagi Perel, Arun Jambulapati, and Kevin Tian for helpful discussions about related work; Taisuke Yasuda for his generous help discussing results in $\ell_p$ sensitivity sampling; and our NeurIPS reviewers for their time, effort, and many constructive suggestions. Swati gratefully acknowledges funding for her student researcher position from Google Research and research

assistantship at UW from the NSF award CCF-1749609 (via Yin Tat Lee). Part of this work was done while D. Woodruff was at Google Research. D. Woodruff also acknowledges support from a Simons Investigator Award.

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

# A Omitted proofs: general technical results

**Fact A.1** (Crude sensitivity approximation via leverage scores)**.** *Given a matrix* $\mathbf{A} \in \mathbb{R}^{n \times d}$*, let* $\boldsymbol{\sigma}_1(\mathbf{a}_i)$ *denote the* $i^{\text{th}}$ $\ell_1$ *sensitivity of* $\mathbf{A}$ *with respect to* $\mathbf{A}$*, and let* $\tau_i(\mathbf{A})$ *denote its* $i^{\text{th}}$ *leverage score with respect to* $\mathbf{A}$*. Then we have* $\sqrt{\frac{\tau_i(\mathbf{A})}{n}} \leq \boldsymbol{\sigma}_1(\mathbf{a}_i) \leq \sqrt{\tau_i(\mathbf{A})}$*.*

*Proof.* The proof of this claim follows from a simple application of standard norm inequalities, and we present one here for completeness. For any $\mathbf{u} \in \mathbb{R}^n$, we have $\|\mathbf{u}\|_2 \leq \|\mathbf{u}\|_1 \leq \sqrt{n}\|\mathbf{u}\|_2$. Therefore, for any $\mathbf{x} \in \mathbb{R}^d$, we have

$$\frac{|\mathbf{x}^\top \mathbf{a}_i|}{\sqrt{n}\|\mathbf{Ax}\|_2} \leq \frac{|\mathbf{x}^\top \mathbf{a}_i|}{\|\mathbf{Ax}\|_1} \leq \frac{|\mathbf{x}^\top \mathbf{a}_i|}{\|\mathbf{Ax}\|_2}. \tag{A.1}$$

Let $\mathbf{x}_\tau$ be the vector that realizes the $i^{\text{th}}$ leverage score, and $\mathbf{x}_\sigma$ be the vector that realizes the $i^{\text{th}}$ $\ell_1$ sensitivity (i.e. they are the maximizers of sensitivity). Then, we may conclude from Inequality (A.1) that

$$\sqrt{\frac{|\mathbf{x}_\tau^\top \mathbf{a}_i|^2}{n\|\mathbf{Ax}_\tau\|_2^2}} \leq \frac{|\mathbf{x}_\tau^\top \mathbf{a}_i|}{\|\mathbf{Ax}_\tau\|_1} \leq \frac{|\mathbf{x}_\sigma^\top \mathbf{a}_i|}{\|\mathbf{Ax}_\sigma\|_1} \leq \sqrt{\frac{|\mathbf{x}_\sigma^\top \mathbf{a}_i|^2}{\|\mathbf{Ax}_\sigma\|_2^2}} \leq \sqrt{\frac{|\mathbf{x}_\tau^\top \mathbf{a}_i|^2}{\|\mathbf{Ax}_\tau\|_2^2}},$$

which gives the claimed result. $\qquad\square$

**Fact A.2.** *Given an* $n \times d$ *matrix, the cost of computing* $k$ *of its* $\ell_1$ *sensitivities is* $\widetilde{O}(\mathbf{nnz}(\mathbf{A}) + k \cdot d^\omega)$*.*

*Proof.* Given an $n \times d$ matrix $\mathbf{A}$, we first compute $\mathbf{SA}$, its $\ell_1$ subspace embedding of size $O(d) \times d$, and compute the $i^{\text{th}}$ sensitivity of $\mathbf{A}$ as follows.

$$\boldsymbol{\sigma}_1^{\mathbf{A}}(\mathbf{a}_i) = \max_{\mathbf{x} \in \mathbb{R}^d} \frac{|\mathbf{a}_i^\top \mathbf{x}|}{\|\mathbf{Ax}\|_1} = \max_{\mathbf{x} \in \mathbb{R}^d} \frac{|\mathbf{a}_i^\top \mathbf{x}|}{\Theta(\|\mathbf{SAx}\|_1)} \approx_{O(1)} \boldsymbol{\sigma}_1^{\mathbf{SA}}(\mathbf{a}_i),$$

where the second step is by the definition of $\ell_1$ subspace embedding (cf. Definition 2.3). To compute $\boldsymbol{\sigma}_1^{\mathbf{SA}}(\mathbf{a}_i)$, we need to compute $\min_{\mathbf{a}_i^\top \mathbf{x}=1} \|\mathbf{SAx}\|_1$, which, by introducing a new variable for each of the rows of $\mathbf{SAx}$, can be transformed into a linear program with $O(d)$ variables and $d$ constraints. To see the claim on runtime, the cost of computing the subspace embedding is $\mathbf{nnz}(\mathbf{A})$. Having computed this once, we can then solve $k$ linear programs, each at cost $d^\omega$ (which is the current fastest LP solver [83]). Putting together these two costs gives the claimed runtime. $\qquad\square$

**Lemma A.3.** *Given positive numbers* $a_1, a_2, \ldots, a_m$*, let* $r := \frac{\max_{i \in [m]} a_i}{\min_{i \in [m]} a_i}$ *and* $A^{(true)} := \sum_{i \in [m]} a_i$*. Suppose we sample, uniformly at random with replacement, a set* $S \in [m]$ *of these numbers, and let* $A^{(est)} := \frac{m}{|S|} \cdot \sum_{i:a_i \in S} a_i$*. Then, if* $|S| \geq 10 \left( \frac{r(1+\gamma)}{\gamma^2} \log \left( \frac{1}{\delta} \right) \right)$ *for a large enough absolute constant* $C$*, we can ensure with at least a probability of* $1 - \delta$*, that* $|A^{(est)} - A^{(true)}| \leq \gamma A^{(true)}$*.*

*Proof.* By construction, the expected value of a sample $a_i$ in $S$ is $\frac{1}{m}A^{(true)}$. Denoting by $\mathcal{P}$ the uniform distribution over $a_1, a_2, \ldots, a_m$, and let $\sigma(\mathcal{P})$ be the standard deviation of this distribution $\mathcal{P}$. Then, we can apply Bernstein's inequality [84, Theorem 2.8.1] to see that the absolute error $|A^{(est)} - A^{(true)}|$ satisfies the following guarantee for all $t > 0$:

$$\Pr\left\{ |A^{(est)} - A^{(true)}| \geq t \right\} = \Pr\left\{ \left| \sum_{i:a_i \in S} \left( a_i - \tfrac{1}{m}A^{(true)} \right) \right| \geq t \cdot \frac{|S|}{m} \right\}$$

$$\leq e^{-\left\{ \frac{\frac{1}{2}t^2 \cdot \frac{|S|^2}{m^2}}{|S| \cdot \sigma(\mathcal{P})^2 + \frac{1}{3}t \cdot \frac{|S|}{m} \cdot \max_{i \in [m]} a_i} \right\}}$$

$$= e^{-\left\{ \frac{\frac{1}{2}t^2 \cdot |S|}{m^2 \cdot \sigma(\mathcal{P})^2 + \frac{1}{3} \cdot m \cdot t \cdot \max_{i \in [m]} a_i} \right\}}. \tag{A.2}$$

Since each of the $a_i$s is positive, we note that

$$\sigma(\mathcal{P})^2 = \frac{1}{m}\sum_{i\in[m]}\left(a_i - \frac{A^{(\text{true})}}{m}\right)^2 \leq \frac{1}{m}\sum_{i\in[m]}a_i^2 \leq \frac{1}{m}\cdot\max_{i\in[m]}a_i\cdot\sum_{i\in[m]}a_i = \frac{1}{m}\cdot\max_{i\in[m]}a_i\cdot A^{(\text{true})}.$$

(A.3)

Combining Inequality (A.2) and Inequality (A.3) for $t = \gamma A^{(\text{true})}$ implies that

$$\Pr\left\{|A^{(\text{est})} - A^{(\text{true})}| \geq \gamma A^{(\text{true})}\right\} \leq e^{-\left\{\frac{\frac{1}{2}\gamma^2 A^{(\text{true})}|S|}{m(1+\frac{1}{3}\cdot\gamma)\cdot\max_{i\in[m]}a_i}\right\}} \leq \delta$$

(A.4)

holds for the choice of $|S| \geq 10\left(\frac{m}{\gamma^2}(1+\gamma)\frac{\max_{i\in[m]}a_i}{A^{(\text{true})}}\log\left(\frac{1}{\delta}\right)\right)$ yields the claimed error guarantee. $\square$

# B   Omitted proofs: $\ell_1$ sensitivities

## B.1   Estimating all $\ell_1$ sensitivities

**Theorem 3.1.** *Given a full-rank matrix $\mathbf{A} \in \mathbb{R}^{n\times d}$ and an approximation factor $1 < \alpha \ll n$, let $\boldsymbol{\sigma}_1(\mathbf{a}_i)$ be the $\ell_1$ sensitivity of the $i^{\text{th}}$ row of $\mathbf{A}$. Then there exists an algorithm that, in time $\widetilde{O}\left(\mathbf{nnz}(\mathbf{A}) + \frac{n}{\alpha}\cdot d^\omega\right)$, returns $\widetilde{\boldsymbol{\sigma}} \in \mathbb{R}^n_{\geq 0}$ such that with high probability, for each $i \in [n]$,*

$$\boldsymbol{\sigma}_1(\mathbf{a}_i) \leq \widetilde{\boldsymbol{\sigma}}_i \leq O(\boldsymbol{\sigma}_1(\mathbf{a}_i) + \tfrac{\alpha}{n}\mathfrak{S}_1(\mathbf{A})).$$

(3.1)

*Proof.* Note that in Line 2 of Algorithm 1, the rows of $\mathbf{A}$ are partitioned into randomly created $n/\alpha$ blocks. Suppose $\mathbf{a}_i$, the $i^{\text{th}}$ row of $\mathbf{A}$, falls into the block $\mathbf{B}_\ell$. Recall, that in Line 5, the rows from $\mathbf{B}_\ell$ are mapped to those in $\mathbf{P}$ with row indices in the set $J$. Then, we compute the $i^{\text{th}}$ $\ell_1$ sensitivity estimate as $\widetilde{\boldsymbol{\sigma}}_i = \max_{j\in J}\boldsymbol{\sigma}_1^{\mathbf{SA}}(\mathbf{p}_j)$. We observe that for all $\mathbf{x} \in \mathbb{R}^d$,

$$\|\mathbf{SAx}\|_1 = \Theta(\|\mathbf{Ax}\|_1).$$

(B.1)

We use Equation (B.1) below to establish the claimed bounds. Let $\mathbf{x}^\star = \arg\max_{\mathbf{x}\in\mathbb{R}^d}\frac{|\mathbf{a}_i^\top\mathbf{x}|}{\|\mathbf{Ax}\|_1}$ be the vector that realizes the $i^{\text{th}}$ $\ell_1$ sensitivity of $\mathbf{A}$. Further, let the $j^{\text{th}}$ row of $\mathbf{P}$ be $\mathbf{r}_k^{(\ell)}\mathbf{B}_\ell$. Then, with a probability of at least $1/2$, we have

$$\boldsymbol{\sigma}_1^{\mathbf{SA}}(\mathbf{p}_j) = \max_{\mathbf{x}\in\mathbb{R}^d}\frac{|\mathbf{r}_k^{(\ell)}\mathbf{B}_\ell\mathbf{x}|}{\|\mathbf{SAx}\|_1} \geq \frac{|\mathbf{r}_k^{(\ell)}\mathbf{B}_\ell\mathbf{x}^\star|}{\|\mathbf{SAx}^\star\|_1} = \Theta(1)\frac{|\mathbf{r}_k^{(\ell)}\mathbf{B}_\ell\mathbf{x}^\star|}{\|\mathbf{Ax}^\star\|_1} \geq \Theta(1)\frac{|\mathbf{a}_i^\top\mathbf{x}^\star|}{\|\mathbf{Ax}^\star\|_1} = \Theta(1)\boldsymbol{\sigma}_1(\mathbf{a}_i),$$

(B.2)

where the first step is by definition of $\boldsymbol{\sigma}_1^{\mathbf{SA}}(\mathbf{p}_j)$, the second is by evaluating the function being maximized at a specific choice of $\mathbf{x}$, the third step is by applying $\|\mathbf{SAx}\|_1 = \Theta(\|\mathbf{Ax}\|_1)$ from Equation (B.1), and the final step is by definition of $\mathbf{x}^\star$ and $\boldsymbol{\sigma}_1(\mathbf{a}_i)$. For the fourth step, we use the fact that $|\mathbf{r}_k^\ell\mathbf{B}_\ell\mathbf{x}^\star| = |\mathbf{r}_{k,i}^\ell\mathbf{a}_i^\top\mathbf{x}^\star + \sum_{j\neq i}\mathbf{r}_{k,j}^\ell(\mathbf{B}_\ell\mathbf{x}^\star)_j| \geq |\mathbf{a}_i^\top\mathbf{x}^\star|$ with a probability of at least $1/2$ since the vector $\mathbf{r}_k^\ell$ has coordinates that are $+1$ or $-1$ with equal probability. By a union bound over the $|J|$ independent rows that block $\mathbf{B}_\ell$ is mapped to, we establish the claimed lower bound in Inequality (B.2) with a probability of at least $0.9$. To show an upper bound on $\boldsymbol{\sigma}_1^{\mathbf{SA}}(\mathbf{p}_j)$, we observe that

$$\boldsymbol{\sigma}_1^{\mathbf{SA}}(\mathbf{p}_j) = \max_{\mathbf{x}\in\mathbb{R}^d}\frac{|\mathbf{r}_k^{(\ell)}\mathbf{B}_\ell\mathbf{x}|}{\|\mathbf{SAx}\|_1} \leq \max_{\mathbf{x}\in\mathbb{R}^d}\frac{\|\mathbf{B}_\ell\mathbf{x}\|_1}{\|\mathbf{SAx}\|_1} = \Theta(1)\max_{\mathbf{x}\in\mathbb{R}^d}\frac{\|\mathbf{B}_\ell\mathbf{x}\|_1}{\|\mathbf{Ax}\|_1} \leq \Theta(1)\sum_{j:\mathbf{a}_j\in\mathbf{B}_\ell}\max_{\mathbf{x}\in\mathbb{R}^d}\frac{|\mathbf{a}_j^\top\mathbf{x}|}{\|\mathbf{Ax}\|_1},$$

(B.3)

where the second step is by Hölder inequality and the fact that the entries of $\mathbf{r}_k^{(\ell)}$ are all bounded between $-1$ and $+1$, and the third step is by Equation (B.1). The final term of the above chain of inequalities may be rewritten as $\Theta(1)\sum_{j:\mathbf{a}_j\in\mathbf{B}_\ell}s_j(\mathbf{A})$, by the definition of $\mathbf{B}_\ell$. Because $\mathbf{B}_\ell$ is a group of $\alpha$ rows selected uniformly at random out of $n$ rows and contains the row $\mathbf{a}_i$, we have:

$$\mathbb{E}\left\{\sum_{\substack{j:\mathbf{a}_j\in\mathbf{B}_\ell,\\j\neq i}}\boldsymbol{\sigma}_1(\mathbf{a}_j)\right\} = \frac{\alpha-1}{n-1}\sum_{j\neq i}\boldsymbol{\sigma}_1(\mathbf{a}_j).$$

Therefore, Markov inequality gives us that with a probability of at least 0.9, we have

$$\boldsymbol{\sigma}_1^{\mathbf{SA}}(\mathbf{p}_j) \leq \boldsymbol{\sigma}_1(\mathbf{a}_j) + \mathfrak{S}_1(\mathbf{A})O(\tfrac{\alpha}{n}). \tag{B.4}$$

The median trick then establishes the bounds in Inequality (B.2) and Inequality (B.4) with high probability. The claimed runtime is obtained by the cost of constructing $\mathbf{SA}$ plus $O(n/\alpha)$ computations of $\ell_1$ sensitivities with respect to $\mathbf{SA}$, for which we may use Fact 2.4. $\qquad\square$

## B.2 Estimating the sum of $\ell_1$ sensitivities

In this section, we present a randomized algorithm that approximates the total sensitivity $\mathfrak{S}_1(\mathbf{A})$, up to a $\gamma$-multiplicative approximation for some $\gamma \in (0, 1)$ that is not too small. This algorithm is significantly less general than Algorithm 2 (which holds for all $p \geq 1$) but it uses extremely simple facts about leverage scores and what we think is a new recursion technique in this context, which we believe could be of potential independent interest.

**Theorem B.1.** *Given a matrix $\mathbf{A} \in \mathbb{R}^{n \times d}$ and an approximation factor $\gamma \in (0, 1)$ such that $\gamma \geq \Omega\left(\frac{1}{n^3}\right)$, there exists an algorithm, which in time $\widetilde{O}\left(\mathbf{nnz}(\mathbf{A}) + \frac{d^\omega}{\gamma^2} \cdot \max\left(\sqrt{d}, \frac{1}{\gamma^2}\right)\right)$, returns a positive scalar $\widehat{s}$ such that, with a probability of $0.99$, we have*

$$\mathfrak{S}_1(\mathbf{A}) \leq \widehat{s} \leq (1 + O(\gamma))\mathfrak{S}_1(\mathbf{A}).$$

---

**Algorithm 4** Approximating the Sum of $\ell_1$-Sensitivities

---

**Input:** Matrix $\mathbf{A} \in \mathbb{R}^{n \times d}$ and approximation factor $\gamma \in (0, 1)$
**Output:** A positive scalar that satisfies, with probability $0.99$, that

$$\mathfrak{S}_1(\mathbf{A}) \leq \widehat{s} \leq (1 + O(\gamma))\mathfrak{S}_1(\mathbf{A})$$

1: Compute $\mathbf{SA} \in \mathbb{R}^{O(d) \times d}$, a $1/2$-approximate $\ell_1$ subspace embedding of $\mathbf{A}$ (cf. Definition 2.3)
2: Set $\rho = O(\gamma/\log(\log(n + d)))$. Compute $\mathbf{S'A}$, a $\rho$-approximate $\ell_1$ subspace embedding of $\mathbf{A}$
3: Construct a matrix $\widehat{\mathbf{A}}$ comprising only those rows of $\mathbf{A}$ with leverage scores at least $1/n^{10}$
4: Let $s$ be the output of Algorithm 5 with inputs $\widehat{\mathbf{A}}$, $\mathbf{SA}$, $\mathbf{S'A}$, $n$, and $\rho$
5: Return $\widehat{s} := (1 + \gamma)\left(s + \frac{1}{n^5}(|\mathbf{A}| - |\widehat{\mathbf{A}}|)\right)$

---

### B.2.1 Proof sketch.

We achieve our guarantee via Algorithm 4 and Algorithm 5, which are based on the following key principles. The first is Fact 2.2: the $i^{\text{th}}$ sensitivity and $i^{\text{th}}$ leverage score of $\mathbf{A} \in \mathbb{R}^{n \times d}$ satisfy $\sqrt{\frac{\tau_i(\mathbf{A})}{n}} \leq \boldsymbol{\sigma}_1(\mathbf{a}_i) \leq \sqrt{\tau_i(\mathbf{A})}$. The second idea, derived from Bernstein's inequality (and formalized in Lemma A.3), is: the sum of a set of positive numbers with values bounded between $a$ and $b$ can be approximated to a $\gamma$-multiplicative factor using $\widetilde{O}((b/a)\gamma^{-2})$ uniformly sampled (and appropriately scaled) samples. The third principle is that $\boldsymbol{\sigma}_1(\mathbf{a}_i) \approx \boldsymbol{\sigma}_1^{\mathbf{SA}}(\mathbf{a}_i)$ when $\mathbf{SA}$ is a constant-approximation $\ell_1$ subspace embedding of $\mathbf{A}$.

Based on these ideas, we first divide the rows of $\mathbf{A}$ into $\mathcal{B} = \Theta(\log n)$ submatrices based on which of the $\mathcal{B}$ buckets $[1/2, 1], [1/4, 1/2], [1/8, 1/4]$, etc. their leverage scores (computed with respect to $\mathbf{A}$) fall into. Since each of these $\mathcal{B}$ submatrices comprises rows with leverage scores in the range $[a, 2a]$ for some $a$, per Fact 2.2, we have $\sqrt{\frac{a}{n}} \leq \boldsymbol{\sigma}_1(\mathbf{a}_i) \leq \sqrt{2a}$. Therefore, by Lemma A.3, the total sensitivity of this submatrix may be approximated by the appropriately scaled total sensitivity of $O(\sqrt{n})$ of its uniformly sampled rows. Our design choice to split rows based on leverage scores is based on the fact that leverage scores (unlike sensitivities) are efficiently computable (using [60]). Computing the total sensitivity of an $\widetilde{O}(\sqrt{n})$-sized matrix thus costs $\widetilde{O}(\mathbf{nnz}(\mathbf{A}) + \sqrt{n} \cdot d^\omega)$ under this scheme.

Applying the idea sketched above recursively further reduces the cost. Suppose, after dividing the input matrix $\mathbf{A}$ into $\mathcal{B}$ submatrices $\mathbf{M}_i$ and subsampling $\widetilde{\mathbf{M}}_i$ from these matrices, we try to recurse on each of these $\mathcal{B}$ submatrices $\widetilde{\mathbf{M}}_i$. A key requirement for this idea to work is that the sum of the $\ell_1$ sensitivities of the rows of $\widetilde{\mathbf{M}}_i$ in isolation be close enough to the true sum of $\ell_1$ sensitivities of

---

**Algorithm 5** Subroutine for Approximating the Sum of $\ell_1$-Sensitivities

---

**Input:** Matrices $\mathbf{M} \in \mathbb{R}^{r \times d}$, $\mathbf{SA} \in \mathbb{R}^{O(d) \times d}$, $\mathbf{S'A} \in \mathbb{R}^{O(d) \times d}$, scalars $n$ and $\rho \in (0,1)$
**Output:** A positive scalar $\widehat{s}$ that satisfies, with probability 0.99, that

$$\mathfrak{S}_1^{\mathbf{S'A}}(\mathbf{M}) \leq \widehat{s} \leq (1 + O(\rho))\mathfrak{S}_1^{\mathbf{S'A}}(\mathbf{M}).$$

1: Set $\mathcal{B} = \Theta(\log n)$, $D = O(\log(\log(n + d)))$, $\delta = \frac{0.01}{\mathcal{B}^D}$ and $b = \widetilde{O}\left(\frac{D^4}{\gamma^2}\max\left(\frac{D^4}{\gamma^2}, \sqrt{d}\right)\right)$
2: **if** the number of rows of $\mathbf{M}$ is at most $b$ **then**
3:      Let $\mathfrak{S}_1^{\mathbf{S'A}}(\mathbf{M})$ be the sum of $\ell_1$ sensitivities of the rows of $\mathbf{M}$ with respect to $\mathbf{S'A}$
4:      Using Fact 2.4 on $\mathbf{M}$, compute $\widehat{s}$ such that $\mathfrak{S}_1^{\mathbf{S'A}}(\mathbf{M}) \leq \widehat{s} \leq (1 + O(\rho))\mathfrak{S}_1^{\mathbf{S'A}}(\mathbf{M})$
5:      Return $\widehat{s}$
6: **else**
7:      Construct the matrix $\mathbf{C} := \begin{bmatrix} \mathbf{M} \\ \mathbf{SA} \end{bmatrix}$. Set $r = O(\sqrt{|\mathbf{C}|}(1 + \rho)\rho^{-2}\log(\delta^{-1}))$.
8:      Compute $\tau(\mathbf{C})$, the vector of leverage scores of $\mathbf{C}$
9:      Divide $\left[\frac{1}{n^{20}}, 1\right]$ into $\mathcal{B}$ geometrically decreasing sub-intervals $[1/2, 1], [1/4, 1/2], \ldots,$ etc.
10:      **for** each bucket $i \in [\mathcal{B}]$ created in the previous step **do**
11:          Construct matrix $\mathbf{M}_i$ with those rows of $\mathbf{M}$ whose leverage scores fall in the $i^{\text{th}}$ bucket
12:          Construct matrix $\widetilde{\mathbf{M}}_i$ composed of $r$ uniformly sampled (with replacement) rows of $\mathbf{M}_i$
13:          Compute $\widetilde{s}_i$, the output of Algorithm 5 with inputs $\widetilde{\mathbf{M}}_i$, $\mathbf{SA}$, $\mathbf{S'A}$, $n$, and $\rho$.
14:      **end for**
15:      Return $\widehat{s} := (1 + \rho) \sum_{i \in [\mathcal{B}]} \frac{|\mathbf{M}_i|}{|\widetilde{\mathbf{M}}_i|} \widetilde{s}_i$
16: **end if**

---

those rows with respect to the original matrix $\mathbf{A}$. In general, *this is not true*. However, we can meet our requirement using the $\ell_1$ subspace embedding $\mathbf{SA}$ of the (original) matrix $\mathbf{A}$. Specifically, if we vertically concatenate a matrix $\widetilde{\mathbf{M}}_i$ with $\mathbf{SA}$ and call this matrix $\mathbf{C}$, then for each row $\widetilde{\mathbf{M}}_i[j] \in \widetilde{\mathbf{M}}_i$, we have $\boldsymbol{\sigma}_1^{\mathbf{C}}(\widetilde{\mathbf{M}}_i[j]) \approx \boldsymbol{\sigma}_1^{\mathbf{A}}(\widetilde{\mathbf{M}}_i[j])$. Further, $|\mathbf{C}| \leq O(\sqrt{n})$, and so in the step subsampling the rows of $\widetilde{\mathbf{M}}_i$, we choose $O(n^{1/4})$ samples. Recursing this exponentially decreases the row dimension of the input matrix at each level, with $(\log n)^k$ matrices at the $k^{\text{th}}$ level of recursion.

### B.2.2    Helper lemmas for proving Theorem B.1

For our formal proof, we need Definition B.2, Fact B.3, and Definition B.4.

**Definition B.2** (Notation for recursion tree of Algorithm 5)**.** *Recall that $\mathcal{B} = \Theta(\log(n))$ as in Line 1 of Algorithm 5. Then we have the following notation corresponding to Algorithm 5.*

▸ *We define a rooted $\mathcal{B}$-ary tree $\mathcal{T}$ corresponding to the execution of Algorithm 5.*
▸ *We use $\mathcal{T}_j$ to denote the subtree starting from the root node with all the nodes up to and including those at the $j^{\text{th}}$ level. We denote the $i^{\text{th}}$ node at the $j^{\text{th}}$ level by $\mathcal{T}_{(j,i)}$. Thus, every node is uniquely specified by its level in the tree and its index within that level.*
▸ *The root node $\mathcal{T}_{(1,1)}$ at level $1$ corresponds to the first call to Algorithm 5 from Algorithm 4.*
▸ *For the node $\mathcal{T}_{(j,i)}$, we denote by $\mathbf{M}^{(j,i)}$ the input matrix to the corresponding recursive call; note that all the other inputs are global and remain the same throughout the recursive call.*
▸ *Analogously, we use $\widehat{s}_{(j,i)}$ to denote our estimate of $\mathfrak{S}_1^{\mathbf{S'A}}(\mathbf{M}^{(j,i)})$, the sum of $\ell_1$ sensitivities of rows of the matrix $\mathbf{M}^{(j,i)}$ in the node $\mathcal{T}_{(j,i)}$, defined with respect to the sketched matrix $\mathbf{S'A}$.*

**Fact B.3.** *If event $A$ happens with probability $1 - \tau_1$ and event $B$ happens with probability $1 - \tau_2$ conditioned on event $A$, then the probability of both events $A$ and $B$ happening is at least $1 - \tau_1 - \tau_2$.*

**Definition B.4.** *We say that a node $\mathcal{N}$ with matrix $\mathbf{M}$ satisfies the $(\widehat{\rho}, \widehat{\delta})$-approximation guarantee if $\widehat{s}$, the output of Algorithm 5 on $\mathbf{M}$, satisfies the following guarantee for the true sum $\mathfrak{S}_1^{\mathbf{S'A}}(\mathbf{M})$ of $\ell_1$ sensitivities of the rows in $\mathbf{M}$:*

$$\widehat{s} \approx_{\widehat{\rho}} \mathfrak{S}_1^{\mathbf{S'A}}(\mathbf{M}) \text{ with a probability at least } 1 - \widehat{\delta}$$

*for some approximation parameter $\widehat{\rho} \in (0, 1)$ and an error probability parameter $\widehat{\delta} \in (0, 1)$.*

**Lemma B.5** (Partitioning of $\mathbf{M}$). *In Algorithm 5, even though we ignore the range $\left[0, \frac{1}{n^{20}}\right]$ when creating the submatrices $\mathbf{M}_1, \mathbf{M}_2, \ldots, \mathbf{M}_{\mathcal{B}}$ from $\mathbf{M}$, every row in $\mathbf{M}$ is part of exactly one of the submatrices $\mathbf{M}_1, \mathbf{M}_2, \ldots, \mathbf{M}_{\mathcal{B}}$. In other words, the matrix $\mathbf{M}$ is partitioned into the matrices $\mathbf{M}_1, \mathbf{M}_2, \ldots, \mathbf{M}_{\mathcal{B}}$, with no row missed out.*

*Proof.* Our proof strategy is to show that for all row indices $k \in [|\mathbf{M}|]$, each of the leverage scores $\tau_k(\mathbf{C})$ satisfy $\tau_k(\mathbf{C}) \geq \frac{1}{n^{20}}$. Therefore, creating buckets of rows with the range of leverage score starting at $\frac{1}{n^{20}}$ (instead of $0$) does not miss out any rows.

To this end, we note that for every row $\mathbf{a}_k$ in the input $\mathbf{M}$ to Algorithm 5, we have:

$$\sqrt{\frac{1}{n^{11}}} \leq \sqrt{\frac{\tau_k(\mathbf{A})}{|\mathbf{A}|}} \leq \boldsymbol{\sigma}_1^{\mathbf{A}}(\mathbf{a}_k) = \max_{\mathbf{x} \in \mathbb{R}^d} \frac{|\mathbf{x}^\top \mathbf{a}_k|}{\|\mathbf{A}\mathbf{x}\|_1} \leq \max_{\mathbf{x} \in \mathbb{R}^d} 3\frac{|\mathbf{x}^\top \mathbf{a}_k|}{\|\mathbf{C}\mathbf{x}\|_1} = 3\boldsymbol{\sigma}_1^{\mathbf{C}}(\mathbf{a}_k) \leq 3\sqrt{\tau_k(\mathbf{C})}, \tag{B.5}$$

where the first step is because in Line 3 of Algorithm 4, we discard those rows $\mathbf{a}_k$ of $\mathbf{A}$ with leverage scores smaller than $1/n^{10}$ and there are $n$ rows in $\mathbf{A}$, so every row $\mathbf{a}_k$ in the input matrix $\mathbf{M}$ to Algorithm 5 satisfies $\tau_k(\mathbf{A}) \geq \frac{1}{n^{10}}$; the second step is by Fact 2.2 applied to $\mathbf{a}_k$; the third step is by the definition of the $\ell_1$ sensitivity of $\mathbf{a}_k$ with respect to $\mathbf{A}$; the fourth step is by observing that $\|\mathbf{C}\mathbf{x}\|_1 = \|\mathbf{S}\mathbf{A}\mathbf{x}\|_1 + \|\mathbf{M}\mathbf{x}\|_1 \leq 2\|\mathbf{A}\mathbf{x}\|_1 + \|\mathbf{A}\mathbf{x}\|_1 = 3\|\mathbf{A}\mathbf{x}\|_1$ since $\mathbf{S}\mathbf{A}$ is a constant-factor (with the constant assumed $1/2$) $\ell_1$ subspace embedding of $\mathbf{A}$ and because $\mathbf{M}$ is a submatrix of $\mathbf{A}$; the fifth step is by the definition of $\ell_1$ sensitivity of $\mathbf{a}_k$ with respect to $\mathbf{C}$; the final step is by Fact 2.2 applied to $\mathbf{a}_k$ within $\mathbf{C}$. Observing the first and last terms of this inequality chain, we have $\tau_k(\mathbf{C}) \geq \frac{1}{3n^{11}}$ for all $k \in [|\mathbf{C}|]$. This finishes the proof of the claim. $\qquad\square$

**Lemma B.6.** *For the recursion tree $\mathcal{T}$ corresponding to Algorithm 5, let the size $b$ of the input matrices at the leaf nodes be $b = \frac{50D^3 \log(100\mathcal{B})}{\gamma^2} \cdot \left(\frac{2D^3 \log(100\mathcal{B})}{\gamma^2} + \sqrt{d \log(d)}\right)$. Then, the depth of the recursion tree is less than $D := 1 + \log(\log(2n + 2d \log(d)))$.*

*Proof.* For now, let $\beta$ be arbitrary but less than $\beta^2 \leq n + d'$, where $d'$ is the row dimension of the subspace embedding $\mathbf{S}\mathbf{A}$. Then, the size of the input matrix to the nodes of the recursion tree as we increase in depth evolves as

$$\beta\sqrt{d' + n}, \ \beta\sqrt{d' + \beta\sqrt{d' + n}}, \ \beta\sqrt{d' + \beta\sqrt{d' + \beta\sqrt{d' + n}}} \ldots.$$

Thus, this sequence evolves as

$$x_{t+1} = \beta\sqrt{d' + x_t}, \text{ with } x_0 = n + d'.$$

To see the limit of this sequence, we set $x = \beta\sqrt{d' + x}$, and solve for this quadratic equation as

$$x = \frac{\beta^2 + \sqrt{\beta^4 + 4\beta^2 d'}}{2} \leq \beta(\beta + \sqrt{d'}). \tag{B.6}$$

We set the base case to be $b^\star := 7\beta(\beta + \sqrt{d'})$, a constant times larger than this asymptotic limit. We now compute the depth $t^\star$ at which the recursion attains the value $b^\star$ is attained. To do so, we consider a sequence $\{y_t\}$ which overestimates the sequence $\{x_t\}$ of matrix sizes for a big range of $t$. We define $\{y_t\}$ as

$$y_0 = n + d', \text{ and } y_{t+1} = \beta\sqrt{2y_t},$$

and it can be checked that for $t$ such that $y_t \geq d'$, we have $y_{t+1} \geq x_{t+1}$. Moreover, for all $t \geq 1$, $y_t = \beta^{1+1/2+\cdots+1/2^{t-1}}(2y_0)^{1/2^t} = \beta^{2-2^{1-t}}(2y_0)^{1/2^t}$. The limit of this sequence can easily be seen to $\beta^2$. We now split our analysis in two cases based on the limits of these two sequences.

**Case 1:** $\beta^2 \geq d'$. Since the limit of the sequence $\{y_t\}$ is $\beta^2$, it is easy to see that $y_t$ is always greater $d'$ (since $y_t$ is monotonically decreasing as $y_0 \geq \beta^2$). Thus, $x_t$ will always be less than $y_t$ in this case. We will now show that $y_t$ (and thus $x_t$) will be less than $b^\star$ in $t_0 = 2 \log \log(2y_0)$ steps. Solving the inequality for $t$ as follows: $y_t \leq \beta^2(2y_0)^{1/2^t} \leq 4\beta^2 \leq b^\star$, it suffices to ensure that $(2y_0)^{1/2^t} \leq 4$, which happens if $t \geq t_0$. Thus, $t^\star \leq 2 \log \log 2y_0$.

**Case 2:** $\beta^2 < d'$. In this case, the sequence $y_t$ will eventually go below $d'$ and thus the inequality $y_t \geq x_t$ might eventually break down. Let $t = t'$ denote the first time step at which $y_t \leq 5d'$ (which will be finite in this case). Then, by definition, $y_t \geq x_t$ for all $t < t'$. We will now upper bound the value of $t'$ as follows: $y_t \leq \beta^2 (2y_0)^{1/2^t} \leq d'(2y_0)^{1/2^t} \leq 4d'$ ; Thus, it suffices to ensure that $(2y_0)^{1/2^t} \leq 4$, which happens if $t \geq t_0 := 2\log\log(2y_0)$. Hence, $t' \leq t_0$. Moreover, the value of $y_{t'} \leq 5d'$, implies that $x_{t'-1} \leq y_{t'-1} \leq \frac{(5d')^2}{2\beta^2}$. We will now show that $x_{t'+1} \leq b^\star$ using the evolution of $x_{t+1} = \beta\sqrt{d' + x_t}$ as follows: First, $x_{t'} \leq \beta\sqrt{d'} + \beta\sqrt{x_{t'-1}} \leq \beta\sqrt{d'} + 5d' \leq 6d'$; Then, $x_{t'+1} \leq \beta\sqrt{d'} + \beta\sqrt{6d'} \leq 7\beta\sqrt{d'} \leq b^\star$. Thus, $t^\star \leq 1 + 2\log\log 2y_0$.

Therefore, in both of these cases, we have that after at most $D = 1 + \log\log(2n + 2d')$ depth, the base case of $b^\star = 7\beta(\beta + \sqrt{d'})$ is achieved at the leaf node. Recall that the algorithm sets $D = 2\log\log(n + d') + 1$ and $\rho = \gamma/D$ for $d' = d\log(d)$ which implies that $\beta := C\frac{(1+\rho)}{\rho^2}\log(\delta^{-1}) = CD\frac{(D+\gamma)}{\gamma^2}\log(100\mathcal{B}^D) \leq \frac{2D^3\log(100\mathcal{B})}{\gamma^2}$. Since Algorithm 5 sets $b = \frac{50D^3\log(100\mathcal{B})}{\gamma^2} \cdot \left(\frac{2D^3\log(100\mathcal{B})}{\gamma^2} + \sqrt{d\log(d)}\right)$, which is larger than $b^\star$, we may conclude that the choice of $b$ implies that the recursion tree $\mathcal{T}$ of Algorithm 5 has a depth less than $D = 1 + \log(\log(2n + 2d\log(d)))$.

$\square$

**Lemma B.7.** *Suppose the root node $\mathcal{T}_{(1,1)}$ of Algorithm 5 satisfies a $(\widehat{\rho}, \widehat{\delta})$-approximation guarantee as defined in Definition B.4, and denote by $\widehat{s}$ the output of Algorithm 4. Then, with a probability of at least $1 - \widehat{\delta}$, we have*

$$\frac{1}{1+\gamma}\widehat{s} \approx_{\widehat{\rho}+\rho} \mathfrak{S}_1^{\mathbf{A}}(\mathbf{A}).$$

As a result of this lemma, in the final proof of correctness of Algorithm 4 (appearing later as proof of Theorem B.1), it suffices to show that $\widehat{s}$ satisfies a $(\widehat{\rho}, \widehat{\delta})$-approximation guarantee for some appropriate choices of these parameters.

*Proof of Lemma B.7.* Suppose the root node satisfies Definition B.4 with some $(\widehat{\rho}, \widehat{\delta})$-approximation guarantee. This means that there exists an $\widetilde{s}$ such that the output of Algorithm 5 on the input matrix $\widehat{\mathbf{A}}$ satisfies

$$\widetilde{s} \approx_{\widehat{\rho}} \mathfrak{S}_1^{\mathbf{S}'\mathbf{A}}(\widehat{\mathbf{A}}) \text{ with a probability at least } 1 - \widehat{\delta}. \tag{B.7}$$

Since $\mathbf{S}'\mathbf{A}$ is, by design, a $\rho$-approximation $\ell_1$ subspace embedding for $\mathbf{A}$, it means for all $\mathbf{x} \in \mathbb{R}^d$ we have

$$(1 - \rho)\|\mathbf{A}\mathbf{x}\|_1 \leq \|\mathbf{S}'\mathbf{A}\mathbf{x}\|_1 \leq (1 + \rho)\|\mathbf{A}\mathbf{x}\|_1. \tag{B.8}$$

Therefore, for every row $\mathbf{a}_i \in \widehat{\mathbf{A}}$, we have by the definition of $\ell_1$ sensitivity and Inequality (B.8) that the $\ell_1$ sensitivity of $\mathbf{a}_i$ computed with respect to $\mathbf{S}'\mathbf{A}$ is a $\rho$-approximation of the $\ell_1$ sensitivity of $\mathbf{a}_i$ with respect to $\mathbf{A}$:

$$\boldsymbol{\sigma}_1^{\mathbf{S}'\mathbf{A}}(\mathbf{a}_i) = \max_{\mathbf{x}\in\mathbb{R}^d} \frac{|\mathbf{a}_i^\top\mathbf{x}|}{\|\mathbf{S}'\mathbf{A}\mathbf{x}\|_1} \approx_\rho \max_{\mathbf{x}\in\mathbb{R}^d} \frac{|\mathbf{a}_i^\top\mathbf{x}|}{\|\mathbf{A}\mathbf{x}\|_1} = \boldsymbol{\sigma}_1^{\mathbf{A}}(\mathbf{a}_i).$$

Therefore, by linearity, this $\rho$-approximation factor transfers over in connecting the total sensitivity of $\widehat{\mathbf{A}}$ with respect to $\mathbf{S}'\mathbf{A}$ to the total sensitivity of $\widehat{\mathbf{A}}$ with respect to $\mathbf{A}$:

$$\mathfrak{S}_1^{\mathbf{S}'\mathbf{A}}(\widehat{\mathbf{A}}) = \sum_{i\in[|\widehat{\mathbf{A}}|]} \boldsymbol{\sigma}_1^{\mathbf{S}'\mathbf{A}}(\mathbf{a}_i) \approx_\rho \sum_{i\in[|\widehat{\mathbf{A}}|]} \boldsymbol{\sigma}_1^{\mathbf{A}}(\mathbf{a}_i) = \mathfrak{S}_1^{\mathbf{A}}(\widehat{\mathbf{A}}). \tag{B.9}$$

Therefore, chaining Equation (B.7) and Equation (B.9) yields the following approximation guarantee:

$$\widetilde{s} \approx_{\widehat{\rho}+\rho} \mathfrak{S}_1^{\mathbf{A}}(\widehat{\mathbf{A}}) \text{ with a probability at least } 1 - \widehat{\delta}. \tag{B.10}$$

There is no change in the error probability above from Equation (B.7) because Equation (B.9) holds deterministically. Next, since in Line 3 of Algorithm 4, we dropped rows of $\mathbf{A}$ that have leverage scores less than $\frac{1}{n^{10}}$, for each of the dropped rows $\mathbf{a}_i$, we have

$$\boldsymbol{\sigma}_1^{\mathbf{A}}(\mathbf{a}_i) \leq \sqrt{\tau_i(\mathbf{A})} \leq \frac{1}{n^5}. \tag{B.11}$$

Therefore, the quantity we return, $\widehat{s}$, satisfies the following approximation guarantee:

$$\frac{1}{1+\gamma}\widehat{s} := \widetilde{s} + \frac{1}{n^5}(|\mathbf{A}| - |\widehat{\mathbf{A}}|) \geq (1 - (\widehat{\rho} + \rho))\mathfrak{S}_1^{\mathbf{A}}(\widehat{\mathbf{A}}) + \sum_{i:\mathbf{a}_i \notin \widehat{\mathbf{A}}} \boldsymbol{\sigma}_1^{\mathbf{A}}(\mathbf{a}_i) \geq (1 - (\widehat{\rho} + \rho))\mathfrak{S}_1^{\mathbf{A}}(\mathbf{A}),$$
(B.12)

where the first step is by definition of $\widehat{s}$ in Algorithm 4; the second step is by Equation (B.10); the third step is by applying Inequality (B.11) to each of the rows not present in $\widehat{\mathbf{A}}$. In the other direction,

$$\frac{1}{1+\gamma}\widehat{s} = \widetilde{s} + \frac{1}{n^5}(|\mathbf{A}| - |\widehat{\mathbf{A}}|) \leq (1 + (\widehat{\rho} + \rho))\mathfrak{S}_1^{\mathbf{A}}(\widehat{\mathbf{A}}) + \frac{1}{n^4} \leq (1 + (\widehat{\rho} + \rho))\mathfrak{S}_1^{\mathbf{A}}(\mathbf{A}), \quad \text{(B.13)}$$

where the first step is by definition of $\widehat{s}$ in Algorithm 4; the second step is by the fact that $|\mathbf{A}| \leq n$; the final step is by combining the assumption $\rho \geq \frac{1}{n^4}$ and the facts that all sensitivities are non-negative and that the total $\ell_1$ sensitivity is at least one. Combining Inequality (B.12) and Inequality (B.13) finishes the proof. $\square$

**Lemma B.8** (Single-Level Computation in Recursion Tree). *Consider a node $\mathcal{N}$ in the recursion tree corresponding to Algorithm 5, and let $\mathbf{M}$ be the input matrix for $\mathcal{N}$. Further assume that $\mathcal{N}$ is not a leaf node. Denote by $\mathcal{N}_1, \mathcal{N}_2, \ldots, \mathcal{N}_{\mathcal{B}}$ (where $\mathcal{B} = \Theta(\log(n))$ is the number of recursive calls at each node in Algorithm 5) the children of node $\mathcal{N}$, with each node $\mathcal{N}_i$ with the input matrix $\widetilde{\mathbf{M}}_i$ (as constructed in Algorithm 5). Suppose each of these child nodes $\mathcal{N}_i$ satisfies a $(\widehat{\rho}, \widehat{\delta})$-approximation guarantee (cf. Definition B.4). Then $\mathcal{N}$ satisfies a $(\widehat{\rho} + \rho, (\widehat{\delta} + \delta) \cdot \mathcal{B})$-approximation guarantee.*

*Proof.* We first reiterate the steps of Algorithm 5 that are crucial to this proof. As proved in Lemma B.5, Algorithm 5 partitions the rows of the matrix $\mathbf{M}$ in node $\mathcal{N}$ into $\mathcal{B}$ submatrices $\mathbf{M}_1, \mathbf{M}_2, \ldots, \mathbf{M}_{\mathcal{B}}$ based on their leverage scores. Next, for each matrix $\mathbf{M}_i$, we uniformly sample with replacement $O(\sqrt{|\mathbf{C}|}(1 + \rho)\rho^{-2}\log(\delta^{-1}))$ rows and term this set of rows as matrix $\widetilde{\mathbf{M}}_i$, which we then pass to Algorithm 5 recursively in node $\mathcal{N}_i$ (recall that we assume that the node $\mathcal{N}$ is not a leaf node). In other words, each child node $\mathcal{N}_i$ has as its input matrix $\widetilde{\mathbf{M}}_i$.

Since we assume that all the child nodes $\mathcal{N}_1, \mathcal{N}_2, \ldots, \mathcal{N}_{\mathcal{B}}$ satisfy a $(\widehat{\rho}, \widehat{\delta})$-approximation guarantee as in Definition B.4, it means that, for all $i \in [\mathcal{B}]$, the output $\widehat{s}_i$ of Algorithm 5 on node $\mathcal{N}_i$ satisfies the following guarantee:

$$\widehat{s}_i \approx_{\widehat{\rho}} \mathfrak{S}_1^{\mathbf{S}'\mathbf{A}}(\widetilde{\mathbf{M}}_i) \text{ with a probability at least } 1 - \widehat{\delta}. \tag{B.14}$$

Next, we recall that each $\mathbf{M}_i$ is composed of only those rows of $\mathbf{M}$ whose leverage scores (computed with respect to $\mathbf{C}$) lie in the $i^{\text{th}}$ bucket $[2^{-i}, 2^{-i+1}]$. Combining this with Fact 2.2, this implies that the $\ell_1$ sensitivity of the $j^{\text{th}}$ row $\mathbf{M}_i[j]$ of the matrix $\mathbf{M}_i$, computed with respect to $\mathbf{C}$, satisfies

$$\sqrt{\frac{2^{-i}}{|\mathbf{C}|}} \leq \sqrt{\frac{\tau_j^{\mathbf{C}}(\mathbf{M}_i)}{|\mathbf{C}|}} \leq \boldsymbol{\sigma}_1^{\mathbf{C}}(\mathbf{M}_i[j]) \leq \sqrt{\tau_j^{\mathbf{C}}(\mathbf{M}_i)} \leq \sqrt{2^{-i+1}}. \tag{B.15}$$

Additionally, we claim that

$$\frac{1}{3(1+\rho)}\boldsymbol{\sigma}_1^{\mathbf{C}}(\mathbf{M}_i[j]) \leq \boldsymbol{\sigma}_1^{\mathbf{S}'\mathbf{A}}(\mathbf{M}_i[j]) \leq 3(1+\rho)\boldsymbol{\sigma}_1^{\mathbf{C}}(\mathbf{M}_i[j]). \tag{B.16}$$

To see this, we observe that

$$\boldsymbol{\sigma}_1^{\mathbf{C}}(\mathbf{M}_i[j]) = \max_{\mathbf{x} \in \mathbb{R}^d} \frac{|\mathbf{x}^\top \mathbf{M}_i[j]|}{\|\mathbf{C}\mathbf{x}\|_1} \geq \max_{\mathbf{x} \in \mathbb{R}^d} \frac{|\mathbf{x}^\top \mathbf{M}_i[j]|}{3\|\mathbf{A}\mathbf{x}\|_1} \geq \max_{\mathbf{x} \in \mathbb{R}^d} \frac{|\mathbf{x}^\top \mathbf{M}_i[j]|}{3(1+\rho)\|\mathbf{S}'\mathbf{A}\mathbf{x}\|_1} = \frac{\boldsymbol{\sigma}_1^{\mathbf{S}'\mathbf{A}}(\mathbf{M}_i[j])}{3(1+\rho)},$$
(B.17)

where the first step is by the definition of $\ell_1$ sensitivity of $\mathbf{M}_i[j]$ (i.e., the $j^{\text{th}}$ row of matrix $\mathbf{M}_i$) with respect to $\mathbf{C}$; the second step is by the definition of $\mathbf{C}$, the fact that $\mathbf{S}\mathbf{A}$ is a $1/2$-factor $\ell_1$ subspace embedding of $\mathbf{A}$, and because $\mathbf{M}$ is a subset of rows of $\mathbf{A}$; the third step is because $\mathbf{S}'\mathbf{A}$ is a $\rho$-approximate $\ell_1$ subspace embedding for $\mathbf{A}$, and the final step is by the definition of $\boldsymbol{\sigma}_1^{\mathbf{A}}(\mathbf{M}_i[j])$. In the other direction, we have

$$\boldsymbol{\sigma}_1^{\mathbf{C}}(\mathbf{M}_i[j]) = \max_{\mathbf{x} \in \mathbb{R}^d} \frac{|\mathbf{x}^\top \mathbf{M}_i[j]|}{\|\mathbf{C}\mathbf{x}\|_1} \leq \max_{\mathbf{x} \in \mathbb{R}^d} \frac{|\mathbf{x}^\top \mathbf{M}_i[j]|}{0.5\|\mathbf{A}\mathbf{x}\|_1} \leq \max_{\mathbf{x} \in \mathbb{R}^d} \frac{(1+\rho)|\mathbf{x}^\top \mathbf{M}_i[j]|}{0.5\|\mathbf{S}'\mathbf{A}\mathbf{x}\|_1} \leq 3(1+\rho)\boldsymbol{\sigma}_1^{\mathbf{S}'\mathbf{A}}(\mathbf{M}_i[j]),$$
(B.18)

where the second step is by the definition of $\mathbf{C}$ as a vertical concatenation of $\mathbf{SA}$ and $\mathbf{M}$ and further using that $\mathbf{SA}$ is a constant factor $\ell_1$ subspace embedding of $\mathbf{A}$ and dropping the non-negative term $\|\mathbf{Mx}\|_1$. Combining Inequality (B.15) and Inequality (B.16), we see that for all rows $j \in [|\mathbf{M}_i|]$, we have

$$\frac{1}{3(1+\rho)}\sqrt{\frac{2^{-i}}{|\mathbf{C}|}} \leq \boldsymbol{\sigma}_1^{\mathbf{S'A}}(\mathbf{M}_i[j]) \leq 3(1+\rho)\sqrt{2^{-i+1}}. \tag{B.19}$$

Since Inequality (B.19) shows upper and lower bounds on each sensitivity $\boldsymbol{\sigma}_1^{\mathbf{S'A}}(\mathbf{M}_i)$ of the matrix $\mathbf{M}_i$, we may use Lemma A.3 to approximate the sum of these sensivities. In particular, by Lemma A.3, we may construct a matrix $\widetilde{\mathbf{M}}_i$ composed of $\frac{20\sqrt{|C|}(1+2\rho)\log(\delta^{-1})}{\rho^2}$ rows of $\mathbf{M}_i$ sampled uniformly at random with replacement, and we are guaranteed

$$(1+\rho)\frac{|\mathbf{M}_i|}{|\widetilde{\mathbf{M}}_i|}\mathfrak{S}_1^{\mathbf{S'A}}(\widetilde{\mathbf{M}}_i) \approx_\rho \mathfrak{S}_1^{\mathbf{S'A}}(\mathbf{M}_i) \text{ with a probability at least } 1-\delta. \tag{B.20}$$

Note that the output $\widehat{s}$ of Algorithm 5 when applied to node $\mathcal{N}$ is defined as

$$\widehat{s} := (1+\rho)\sum_{i\in[\mathcal{B}]}\frac{|\mathbf{M}_i|}{|\widetilde{\mathbf{M}}_i|}\widehat{s}_i. \tag{B.21}$$

Therefore, by union bound over the failure probability (cf. Fact B.3), we can combine Equation (B.14) and sum Equation (B.20) over all $i \in \mathcal{B}$ child nodes to conclude that $\widehat{s}$ from Equation (B.21) satisfies a $(\widehat{\rho}+\rho, (\widehat{\delta}+\delta)\mathcal{B})$-approximation guarantee of Definition B.4. $\qquad\square$

### B.2.3   Proof of Theorem B.1

*Proof of Theorem B.1.* We first sketch our strategy to prove Theorem B.1's correctness guarantee. Essentially, our goal is to establish Definition B.4 for the root node $\mathcal{T}_{(1,1)}$. That this goal suffices to prove the theorem is justified in Lemma B.7. We now show that the node $\mathcal{T}_{(1,1)}$ satisfies a $(\widehat{\rho}, \widehat{\delta})$-approximation guarantee for some $\widehat{\rho}$ and $\widehat{\delta}$. We use the method of induction.

**Correctness guarantee of Theorem B.1.** We know that the leaf nodes all satisfy the $(\rho, 0)$-approximation guarantee in Definition B.4 since at this level, the base case is triggered, where we compute a $\rho$-approximation estimate of the total $\ell_1$ sensitivity with respect to the subspace embedding $\mathbf{S'A}$. Having established this approximation guarantee at the leaf nodes, we can use Lemma B.8 to inductively propagate the property of $(\rho, \delta)$-approximation guarantee up the recursion tree $\mathcal{T}$. From Lemma B.6, we have that the recursion tree $\mathcal{T}$ of Algorithm 5 is of depth at most $D = O(\log\log(n+d))$, and the number of child nodes at any given node is at most $\mathcal{B} = \Theta(\log(n))$; we factor these into the approximation guarantee and error probability.

**Base case.** At the leaf nodes of the recursion tree, we directly compute the total sensitivities with respect to $\mathbf{S'A}$. Recall that we denote the depth of the tree by $D$. Then, because $\mathbf{S'A}$ is, by definition, a $\rho$-approximate subspace embedding for $\mathbf{A}$, we have, for each $i$ that indexes a leaf node,

$$\widehat{s}_{(D,i)} \approx_\rho \mathfrak{S}_1^{\mathbf{S'A}}(\mathbf{M}^{(D,i)}) \text{ with a probability of } 1. \tag{B.22}$$

**Induction proof.** Consider the subtree $\mathcal{T}_j$ truncated at (but including) the nodes at level $j$ in the recursion tree $\mathcal{T}$ corresponding to Algorithm 5. Assume the induction hypothesis that all the leaf nodes in the subtree $\mathcal{T}_j$ satisfy Definition B.4 with parameters $\left(\rho\cdot(D+1-j), \delta\cdot\left(\frac{\mathcal{B}^{D-j+1}-\mathcal{B}}{\mathcal{B}-1}\right)\right)$. That is, at the matrix of each such node $i$, we have an estimate $\widehat{s}_{(j,i)}$ of its total sensitivity $\mathfrak{S}_1^{\mathbf{S'A}}(\mathbf{M}^{(j,i)})$ such that

$$\widehat{s}_{(j,i)} \approx_{\rho\cdot(D+1-j)} \mathfrak{S}_1^{\mathbf{S'A}}(\mathbf{M}^{(j,i)}) \text{ with a probability at least } 1 - \delta\cdot\left(\frac{\mathcal{B}^{D-j+1}-\mathcal{B}}{\mathcal{B}-1}\right). \tag{B.23}$$

Then, each of the leaf nodes of the subtree $\mathcal{T}_{j-1}$ (i.e., the subtree that stops one level above $\mathcal{T}_j$) satisfies the assumption in Lemma B.8 since its child nodes are either leaf nodes of the entire recursion tree $\mathcal{T}$ (for which this statement has been shown in Inequality (B.22)) or, if they are not

leaf nodes of the recursion tree, they satisfy Definition B.4 with parameters $\widehat{\rho} = \rho \cdot (D + 1 - j)$ and $\widehat{\delta} = \delta \cdot \left( \frac{\mathcal{B}^{D-j+1} - \mathcal{B}}{\mathcal{B} - 1} \right)$. Therefore, Lemma B.8 implies that each of the leaf nodes of $\mathcal{T}_{j-1}$ satisfies Definition B.4 with approximation parameter $\widehat{\rho}$ and error probability $\widehat{\delta}$ defined as follows:

$$\widehat{\rho} = \rho + \rho \cdot (D + 1 - j) = \rho \cdot (D + 2 - j) \text{ and } \widehat{\delta} = \mathcal{B} \cdot \left( \delta + \delta \cdot \left( \frac{\mathcal{B}^{D-j+1} - \mathcal{B}}{\mathcal{B} - 1} \right) \right) = \delta \cdot \left( \frac{\mathcal{B}^{D-j+2} - \mathcal{B}}{\mathcal{B} - 1} \right).$$

In other words, the outputs $\widehat{s}_{(j-1,k)}$ of Algorithm 5 on the matrices $\mathbf{M}^{(j-1,k)}$ in the leaf nodes of $\mathcal{T}_{j-1}$ each satisfy:

$$\widehat{s}_{(j-1,k)} \approx_{O(\rho \cdot (D+2-j))} \mathfrak{S}_1^{\mathbf{S}'\mathbf{A}}(\mathbf{M}^{(j-1,k)}) \text{ with a probability at least } 1 - \delta \cdot \left( \frac{\mathcal{B}^{D-j+2} - \mathcal{B}}{\mathcal{B} - 1} \right). \tag{B.24}$$

The base case in Inequality (B.22) and the satisfaction of the induction hypothesis in Inequality (B.24) together finish the proof of the induction hypothesis.

**Finishing the proof of correctness.** In light of the above conclusion, for the root node $\mathcal{T}_{(1,1)}$, which is at the level $j = 1$, we may plug in $j = 1$ in Inequality (B.23) and obtain that the output $\widehat{s}_{(1,1)}$ of Algorithm 5 at the root node satisfies

$$\widehat{s}_{(1,1)} \approx_{O(D\rho)} \mathfrak{S}_1^{\mathbf{S}'\mathbf{A}}(\mathbf{M}^{(1,1)}) \text{ with a probability of at least } 1 - \delta \left( \frac{\mathcal{B}^D - \mathcal{B}}{\mathcal{B} - 1} \right). \tag{B.25}$$

In Algorithm 5, we choose the parameters $\delta = \frac{0.01}{\mathcal{B}^D}$ and $\rho = O\left( \frac{\gamma}{D} \right)$. Plugging these parameters into Inequality (B.25), we conclude that the root node satisfies Definition B.4 with a $(\gamma, 0.01)$-approximation guarantee. This finishes the proof of correctness of the claim.

**Proof of runtime.** Let $b$ denote the maximum number of rows of a matrix at a leaf node of $\mathcal{T}$. Further, recall from Fact 2.4 that the cost of computing one $\ell_1$ sensitivity of an $n \times d$ matrix is,

$$L = \widetilde{O}(\mathbf{nnz}(\mathbf{A}) + d^\omega). \tag{B.26}$$

Then as stated in Line 1, the algorithm computes the $\rho$-approximate total sensitivity at the leaf nodes; Fact 2.4, this incurs a cost of

$$\mathcal{C}(b) = \widetilde{O}(\mathbf{nnz}(\mathbf{A}) + b \cdot d^\omega). \tag{B.27}$$

At any other node, the computational cost is the work done at that level plus the total cost of the recursive calls at that level. Let $\mathcal{C}(r)$ denote the runtime of Algorithm 5 when the input is an $r \times d$ matrix. Then,

$$\mathcal{C}(r) = \begin{cases} \mathcal{B} \cdot \mathcal{C}\left( \frac{\sqrt{r}(1+\rho)}{\rho^2} \log(\delta^{-1}) \right) + L & r > b \\ \mathcal{C}(b) & r \leq b \end{cases},$$

where $b$ is the number of rows in the base case, and $L$ is the cost of computing leverage scores and equals $L = \widetilde{O}(\mathbf{nnz}(\mathbf{A}) + d^\omega)$ from Fact 2.1. For the depth of the recursion $D$ (where the top level is $D = 1$), plugging in our choices of $\delta = \frac{0.01}{\mathcal{B}^D}$ and $\rho = O\left( \frac{\gamma}{D} \right)$, this may be expressed as via expanding the recursion as:

$$\mathcal{C}(n) = \mathcal{B}^D \cdot \mathcal{C}(b) + L \cdot \left( \frac{\mathcal{B}^D - 1}{\mathcal{B} - 1} \right) \tag{B.28}$$

for $n \geq b$. We plug into this expression the values of $b$ and $D$ from Lemma B.6, $L$ from Equation (B.26), $\mathcal{C}(b)$ from Equation (B.27), and $\mathcal{B} = \Theta(\log(n))$ to get the claimed runtime. $\qquad\square$

## B.3 Estimating the maximum $\ell_1$ sensitivity

**Theorem 3.7 (Approximating the Maximum of $\ell_1$ Sensitivities).** *Given a matrix $\mathbf{A} \in \mathbb{R}^{n \times d}$, there exists an algorithm, which in time $\widetilde{O}(\mathbf{nnz}(\mathbf{A}) + d^{\omega+1})$, outputs a positive scalar $\widehat{s}$ that satisfies*

$$\Omega(\|\boldsymbol{\sigma}_1(\mathbf{A})\|_\infty) \leq \widehat{s} \leq O(\sqrt{d}\|\boldsymbol{\sigma}_1(\mathbf{A})\|_\infty).$$

*Proof.* First, we have

$$\|\mathbf{S}_1 \mathbf{A}\mathbf{x}\|_1 = \Theta(\|\mathbf{A}\mathbf{x}\|_1). \tag{B.29}$$

We use Equation (B.29) to establish the claimed bounds below. First we set some notation. Define $\mathbf{x}^\star$ and $\mathbf{a}_{i^\star}$ as follows:

$$\mathbf{x}^\star, i^\star = \arg \max_{\mathbf{x} \in \mathbb{R}^d, i \in [|\mathbf{A}|]} \frac{|\mathbf{a}_i^\top \mathbf{x}|}{\|\mathbf{A}\mathbf{x}\|_1} \tag{B.30}$$

Thus, $\mathbf{x}^\star$ is the vector that realizes the maximum sensitivity of the matrix $\mathbf{A}$, and the row $\mathbf{a}_{i^\star}$ is the row of $\mathbf{A}$ with maximum sensitivity with respect to $\mathbf{A}$. Suppose the matrix $\mathbf{S}_\infty \mathbf{A}$ contains the row $\mathbf{a}_{i^\star}$. Then we have

$$\max_{i: \mathbf{c}_i \in \mathbf{S}_\infty \mathbf{A}} \boldsymbol{\sigma}_1(\mathbf{c}_i) = \max_{\mathbf{x} \in \mathbb{R}^d, \mathbf{c}_k \in \mathbf{S}_\infty \mathbf{A}} \frac{|\mathbf{c}_k^\top \mathbf{x}|}{\|\mathbf{S}_1 \mathbf{A}\mathbf{x}\|_1} = \Theta(1) \max_{\mathbf{x} \in \mathbb{R}^d, \mathbf{c}_k \in \mathbf{S}_\infty \mathbf{A}} \frac{|\mathbf{c}_k^\top \mathbf{x}|}{\|\mathbf{A}\mathbf{x}\|_1} = \Theta(1) \max_{\mathbf{x} \in \mathbb{R}^d} \frac{|\mathbf{a}_{i^\star}^\top \mathbf{x}|}{\|\mathbf{A}\mathbf{x}\|_1}, \tag{B.31}$$

where the first step is by definition of $\ell_1$ sensitivity of $\mathbf{S}_\infty \mathbf{A}$ with respect to $\mathbf{S}_1 \mathbf{A}$, the second step is by Equation (B.29), and the third step by noting that the matrix $\mathbf{S}_\infty \mathbf{A}$ is a subset of the rows of $\mathbf{A}$ (which includes $\mathbf{a}_{i^\star}$). By definition of $\boldsymbol{\sigma}_1(\mathbf{a}_{i^\star})$ in Equation (B.30), we have

$$\max_{\mathbf{x} \in \mathbb{R}^d} \frac{|\mathbf{a}_{i^\star}^\top \mathbf{x}|}{\|\mathbf{A}\mathbf{x}\|_1} = \|\boldsymbol{\sigma}_1(\mathbf{A})\|_\infty, . \tag{B.32}$$

Then, combining Equation (B.31) and Inequality (B.32) gives the guarantee in this case. In the other case, suppose $\mathbf{a}_{i^\star}$ is not included in $\mathbf{S}_\infty \mathbf{A}$. Then we observe that the upper bound from the preceding inequalities still holds. For the lower bound, we observe that

$$\|\boldsymbol{\sigma}_1^{\mathbf{S}_1 \mathbf{A}}(\mathbf{S}_\infty \mathbf{A})\|_\infty = \max_{\mathbf{x} \in \mathbb{R}^d, \mathbf{c}_j \in \mathbf{S}_\infty \mathbf{A}} \frac{\mathbf{c}_j^\top \mathbf{x}}{\|\mathbf{S}_1 \mathbf{A}\mathbf{x}\|_1} \geq \max_{\mathbf{x} \in \mathbb{R}^d} \frac{\|\mathbf{S}_\infty \mathbf{A}\mathbf{x}\|_\infty}{\|\mathbf{S}_1 \mathbf{A}\mathbf{x}\|_1} = \Theta(1) \max_{\mathbf{x} \in \mathbb{R}^d} \frac{\|\mathbf{S}_\infty \mathbf{A}\mathbf{x}\|_\infty}{\|\mathbf{A}\mathbf{x}\|_1}, \tag{B.33}$$

where the the second step is by choosing a specific vector in the numerator and the third step uses $\|\mathbf{S}_1 \mathbf{A}\mathbf{x}\|_1 = \Theta(\|\mathbf{A}\mathbf{x}\|_1)$. We further have,

$$\max_{\mathbf{x} \in \mathbb{R}^d} \frac{\|\mathbf{S}_\infty \mathbf{A}\mathbf{x}\|_\infty}{\|\mathbf{A}\mathbf{x}\|_1} \geq \frac{\|\mathbf{S}_\infty \mathbf{A}\mathbf{x}^\star\|_\infty}{\|\mathbf{A}\mathbf{x}^\star\|_1} \geq \frac{\|\mathbf{A}\mathbf{x}^\star\|_\infty}{\sqrt{d}\|\mathbf{A}\mathbf{x}^\star\|_1} \geq \frac{|\mathbf{a}_{i^\star}^\top \mathbf{x}^\star|}{\sqrt{d}\|\mathbf{A}\mathbf{x}^\star\|_1} = \frac{1}{\sqrt{d}} \|\boldsymbol{\sigma}_1(\mathbf{A})\|_\infty, \tag{B.34}$$

where the first step is by choosing $\mathbf{x} = \mathbf{x}^\star$, the second step is by the distortion guarantee of $\ell_\infty$ subspace embedding [35], and the final step is by definition of $\boldsymbol{\sigma}_1(\mathbf{a}_{i^\star})$. Combining Inequality (C.10) and Inequality (C.11) gives the claimed lower bound on $\|\boldsymbol{\sigma}_1^{\mathbf{S}_1 \mathbf{A}}(\mathbf{S}_\infty \mathbf{A})\|_\infty$. The runtime follows from the computational cost and row dimension of $\mathbf{S}_\infty \mathbf{A}$ from [35] and the cost of computing $\ell_1$ sensitivities with respect to $\mathbf{S}_1 \mathbf{A}$ as per Fact 2.4. $\qquad\square$

# C  Omitted proofs: $\ell_p$ sensitivities

We recall the following notation that we use for stating our results in this setting.

**Definition C.1** (Notation for $\ell_p$ Results)**.** *We introduce the notation* $\mathsf{LP}(m, d, p)$ *to denote the cost of approximating one $\ell_p$ sensitivity of an $m \times d$ matrix up to an accuracy of a given constant factor.*

## C.1  Estimating all $\ell_p$ sensitivities

We generalize Algorithm 1 to the case $p \geq 1$ below.

**Theorem C.2** (Approximating all $\ell_p$ sensitivities)**.** *Given a full-rank matrix $\mathbf{A} \in \mathbb{R}^{n \times d}$, an approximation factor $1 < \alpha \ll n$, and a scalar $p \geq 1$, let $\boldsymbol{\sigma}_p(\mathbf{a}_i)$ be the $i^{\text{th}}$ $\ell_p$ sensitivity. Then there exists an algorithm that returns a vector $\widetilde{\boldsymbol{\sigma}} \in \mathbb{R}_{\geq 0}^n$ such that with high probability, for each $i \in [n]$, we have*

$$\Omega(\boldsymbol{\sigma}_p(\mathbf{a}_i)) \leq \widetilde{\boldsymbol{\sigma}}_i \leq O(\alpha^{p-1} \boldsymbol{\sigma}_p(\mathbf{a}_i)) + \frac{\alpha^p}{n} \mathfrak{S}_p(\mathbf{A}). \tag{C.1}$$

*Our algorithm runs in time* $\widetilde{O}\left(\mathbf{nnz}(\mathbf{A}) + \frac{n}{\alpha} \cdot \mathsf{LP}(d^{\max(1, p/2)}, d, p)\right)$ *(cf. Definition 1.2).*

---
**Algorithm 6** Approximating $\ell_p$-Sensitivities: Row-wise Approximation
---

**Inputs:** Matrix $\mathbf{A} \in \mathbb{R}^{n \times d}$, approximation factor $\alpha \in (1, n]$, scalar $p \geq 1$
**Output:** Vector $\widetilde{\boldsymbol{\sigma}} \in \mathbb{R}_{>0}^n$ that satisfies, for each $i \in [n]$, with probability 0.9, that

$$\boldsymbol{\sigma}_p(\mathbf{a}_i) \leq \widetilde{\boldsymbol{\sigma}}_i \leq O(\alpha^{p-1} \boldsymbol{\sigma}_p(\mathbf{a}_i) + \tfrac{\alpha^p}{n} \mathfrak{S}_p(\mathbf{A}))$$

1: Compute, for $\mathbf{A}$, an $\ell_p$ subspace embedding $\mathbf{S}_p \mathbf{A} \in \mathbb{R}^{\widetilde{O}(d^{\max(1,p/2)}) \times d}$
2: Partition $\mathbf{A}$ into $\frac{n}{\alpha}$ blocks $\mathbf{B}_1, \ldots, \mathbf{B}_{n/\alpha}$ each comprising $\alpha$ randomly selected rows.
3: **for** the $\ell^{\text{th}}$ block $\mathbf{B}_\ell$, with $\ell \in [\frac{n}{\alpha}]$ **do**
4:     Sample $\alpha$-dimensional independent Rademacher vectors $\mathbf{r}_1^{(\ell)}, \ldots, \mathbf{r}_{100}^{(\ell)}$
5:     For each $j \in [100]$, compute the row vectors $\mathbf{r}_j^{(\ell)} \mathbf{B}_\ell \in \mathbb{R}^d$
6: **end for**
7: Let $\mathbf{P} \in \mathbb{R}^{100\frac{n}{\alpha} \times d}$ be the matrix of all vectors computed in Line 5. Compute $\boldsymbol{\sigma}_p^{\mathbf{S}_p \mathbf{A}}(\mathbf{P})$ using Definition 1.2
8: **for** each $i \in [n]$ **do**
9:     Denote by $J$ the set of row indices in $\mathbf{P}$ that $\mathbf{a}_i$ is mapped to in Line 5
10:     Set $\widetilde{\boldsymbol{\sigma}}_i = \max_{j \in J}(\boldsymbol{\sigma}_p^{\mathbf{S}_p \mathbf{A}}(\mathbf{p}_j))$
11: **end for**
12: **Return** $\widetilde{\boldsymbol{\sigma}}$

---

*Proof.* Suppose the $i^{\text{th}}$ row of $\mathbf{A}$ falls into the bucket $\mathbf{B}_\ell$. Suppose, further, that the rows from $\mathbf{B}_\ell$ are mapped to those in $\mathbf{P}$ with row indices in the set $J$. Then, Algorithm 6 returns a vector of sensitivity estimates $\widetilde{\boldsymbol{\sigma}} \in \mathbb{R}^n$ with the $i^{\text{th}}$ coordinate defined as $\widetilde{\boldsymbol{\sigma}}_i = \max_{j \in J} \boldsymbol{\sigma}_p(\mathbf{c}_j)$. For any $\mathbf{x} \in \mathbb{R}^d$, we have

$$\|\mathbf{S}_p \mathbf{A} \mathbf{x}\|_p^p \approx \|\mathbf{A} \mathbf{x}\|_p^p. \tag{C.2}$$

We use Equation (C.2) to establish the claimed bounds below. Let $\mathbf{x}^\star = \arg\max_{\mathbf{x} \in \mathbb{R}^d} \frac{|\mathbf{a}_i^\top \mathbf{x}|^p}{\|\mathbf{A} \mathbf{x}\|_p^p}$ be the vector that realizes the $i^{\text{th}}$ $\ell_p$ sensitivity of $\mathbf{A}$. Further, let the $j^{\text{th}}$ row of $\mathbf{P}$ be $\mathbf{r}_k^{(\ell)} \mathbf{B}_\ell$. Then, with probability of at least $1/2$, we have

$$\boldsymbol{\sigma}_p^{\mathbf{S}_p \mathbf{A}}(\mathbf{p}_j) = \max_{\mathbf{x} \in \mathbb{R}^d} \frac{|\mathbf{r}_k^{(\ell)} \mathbf{B}_\ell \mathbf{x}|^p}{\|\mathbf{S}_p \mathbf{A} \mathbf{x}\|_p^p} \geq \frac{|\mathbf{r}_k^{(\ell)} \mathbf{B}_\ell \mathbf{x}^\star|^p}{\|\mathbf{S}_p \mathbf{A} \mathbf{x}^\star\|_p^p} = \Theta(1) \frac{|\mathbf{r}_k^{(\ell)} \mathbf{B}_\ell \mathbf{x}^\star|^p}{\|\mathbf{A} \mathbf{x}^\star\|_p^p} \geq \Theta(1) \frac{|\mathbf{a}_i^\top \mathbf{x}^\star|^p}{\|\mathbf{A} \mathbf{x}^\star\|_p^p} = \Theta(1) \boldsymbol{\sigma}_p(\mathbf{a}_i),$$
$$\tag{C.3}$$

where the first step is by definition of $\boldsymbol{\sigma}_p(\mathbf{p}_j)$, the second is by evaluating the function being maximized at a specific choice of $\mathbf{x}$, the third step is by Equation (C.2), and the final step is by definition of $\mathbf{x}^\star$ and $\boldsymbol{\sigma}_p(\mathbf{a}_i)$. For the fourth step, we use the fact that $|\mathbf{r}_k^{(\ell)} \mathbf{B}_\ell \mathbf{x}^\star|^p = |\mathbf{r}_{k,i}^{(\ell)} \mathbf{a}_i^\top \mathbf{x}^\star + \sum_{j \neq i} \mathbf{r}_{k,j}^{(\ell)} (\mathbf{B} \mathbf{x}^\star)_j|^p \geq |\mathbf{a}_i^\top \mathbf{x}^\star|^p$ with a probability of at least $1/2$ since the vector $\mathbf{r}_k^{(\ell)}$ has coordinates that are $-1$ or $+1$ with equal probability. By a union bound over $j \in J$ independent rows that block $\mathbf{B}_\ell$ is mapped to, we establish the claimed lower bound in Inequality (C.3) with probability at least 0.9. To show an upper bound on $\boldsymbol{\sigma}_p^{\mathbf{S}_p \mathbf{A}}(\mathbf{p}_j)$, we observe that

$$\max_{\mathbf{x} \in \mathbb{R}^d} \frac{|\mathbf{r}_k^{(\ell)} \mathbf{B}_\ell \mathbf{x}|^p}{\|\mathbf{S}_p \mathbf{A} \mathbf{x}\|_p^p} \leq \alpha^{p-1} \max_{\mathbf{x} \in \mathbb{R}^d} \frac{\|\mathbf{B}_\ell \mathbf{x}\|_p^p}{\|\mathbf{S}_p \mathbf{A} \mathbf{x}\|_p^p} \leq \Theta(\alpha^{p-1}) \sum_{j : \mathbf{a}_j \in \mathbf{B}_\ell} \max_{\mathbf{x} \in \mathbb{R}^d} \frac{|\mathbf{a}_j^\top \mathbf{x}|^p}{\|\mathbf{A} \mathbf{x}\|_p^p} = \Theta(\alpha^{p-1}) \sum_{j : \mathbf{a}_j \in \mathbf{B}_\ell} \boldsymbol{\sigma}_p(\mathbf{a}_j),$$
$$\tag{C.4}$$

where the second step is by Equation (C.2) and opening up $\|\mathbf{B}_\ell \mathbf{x}\|_p^p$ in terms of the rows in $\mathbf{B}_\ell$, and the final step is by definition of $\mathbf{B}_\ell$. To see the first step, we observe that

$$|\mathbf{r}_k^{(\ell)} \mathbf{B}_\ell \mathbf{x}|^p \leq \|\mathbf{B}_\ell \mathbf{x}\|_p^p \cdot \|\mathbf{r}_k^{(\ell)}\|_q^p \leq \|\mathbf{B}_\ell \mathbf{x}\|_p^p \cdot \alpha^{p/q} = \|\mathbf{B}_\ell \mathbf{x}\|_p^p \cdot \alpha^{p-1},$$

where $\frac{1}{p} + \frac{1}{q} = 1$, the first step is by Hölder's inequality, the second is because each entry of $\mathbf{r}_k^{(\ell)}$ is either $+1$ or $-1$ and $\mathbf{r}_k^{(\ell)} \in \mathbb{R}^\alpha$, and the third is because $\frac{p}{q} = p - 1$. Because $\mathbf{B}_\ell$ is a group of $\alpha$ rows

selected uniformly at random out of $n$ rows and contains the row $\mathbf{a}_i$, we have:

$$\mathbb{E}\left\{\sum_{j:\mathbf{a}_j\in\mathbf{B}_\ell,j\neq i}\boldsymbol{\sigma}_p(\mathbf{a}_j)\right\} = \frac{\alpha-1}{n-1}\sum_{j\neq i}\boldsymbol{\sigma}_p(\mathbf{a}_i).$$

Therefore, Markov inequality gives us that with a probability of at least $0.9$, we have

$$\boldsymbol{\sigma}_p^{\mathbf{S}_p\mathbf{A}}(\mathbf{p}_j) \leq O(\alpha^{p-1}\boldsymbol{\sigma}_p(\mathbf{a}_i) + \tfrac{\alpha^p}{n}\mathfrak{S}_p(\mathbf{A})). \tag{C.5}$$

The median trick then establishes Inequality (C.4) and Inequality (C.5) with high probability. To establish the runtime, note that we first construct the subspace embedding $\mathbf{S}_p\mathbf{A}$ and then compute the $\ell_p$ sensitivities of $\mathbf{P}$ (with $O(n/\alpha)$ rows) with respect to $\mathbf{S}_p\mathbf{A}$. The size of the subspace embedding $\mathbf{S}_p\mathbf{A}$ is $O(d^{1\vee p/2}\times d)$ (see [75, Table 1]). This completes the runtime claim. $\qquad\square$

## C.2 Estimating the maximum of $\ell_p$ sensitivities

In Section 3.2 and Appendix B.3, we showed our result for estimating the maximum $\ell_1$ sensitivity. In this section, we show how to extend this result to all $p \geq 1$. Algorithm 7 generalizes Algorithm 3.

---

**Algorithm 7** Approximating the Maximum of $\ell_p$-Sensitivities

---

**Input:** Matrix $\mathbf{A} \in \mathbb{R}^{n\times d}$ and $p \geq 1$ (with $p \neq 2$)
**Output:** Scalar $\widehat{s} \in \mathbb{R}_{\geq 0}$ that satisfies that

$$\|\boldsymbol{\sigma}_p(\mathbf{A})\|_\infty \leq \widehat{s} \leq C \cdot d^{p/2} \cdot \|\boldsymbol{\sigma}_p(\mathbf{A})\|_\infty$$

1: Compute, for $\mathbf{A}$, an $\ell_\infty$ subspace embedding $\mathbf{S}_\infty\mathbf{A} \in \mathbb{R}^{O(d\log^2(d))\times d}$ such that $\mathbf{S}_\infty\mathbf{A}$ is a subset of the rows of $\mathbf{A}$ [35]
2: Compute, for $\mathbf{A}$, an $\ell_p$ subspace embedding $\mathbf{S}_p\mathbf{A} \in \begin{cases} \mathbb{R}^{O(d)\times d} & p\in[1,2) \\ \mathbb{R}^{O(d^{p/2})\times d} & p>2 \end{cases}$
3: Return $d^{p/2}\|\boldsymbol{\sigma}_p^{\mathbf{S}_p\mathbf{A}}(\mathbf{S}_\infty\mathbf{A})\|_\infty$

---

**Theorem C.3** (Approximating the maximum of $\ell_p$ sensitivities). *Given a matrix $\mathbf{A} \in \mathbb{R}^{n\times d}$ and $p \geq 1$ (with[3] $p \neq 2$), there exists an algorithm, which outputs a positive scalar $\widehat{s}$ that satisfies*

$$\Omega(\|\boldsymbol{\sigma}_p(\mathbf{A})\|_\infty) \leq \widehat{s} \leq O(d^{p/2}\|\boldsymbol{\sigma}_p(\mathbf{A})\|_\infty).$$

*The runtime of the algorithm is $\widetilde{O}(\mathbf{nnz}(\mathbf{A}) + d^{\omega+1})$ for $p \in [1,2)$ and $\widetilde{O}(\mathbf{nnz}(\mathbf{A}) + d^{p/2} \cdot \mathsf{LP}(O(d^{p/2}),d,p))$ for $p > 2$.*

*Proof.* First, we have

$$\|\mathbf{S}_p\mathbf{A}\mathbf{x}\|_p^p = \Theta(\|\mathbf{A}\mathbf{x}\|_p^p). \tag{C.6}$$

We use Equation (C.6) to establish the claimed bounds below. First we set some notation. Define $\mathbf{x}^\star$ and $\mathbf{a}_{i^\star}$ as follows:

$$\mathbf{x}^\star, i^\star = \arg\max_{\mathbf{x}\in\mathbb{R}^d, i\in[|\mathbf{A}|]}\frac{|\mathbf{a}_i^\top\mathbf{x}|^p}{\|\mathbf{A}\mathbf{x}\|_p^p} \tag{C.7}$$

Thus, $\mathbf{x}^\star$ is the vector that realizes the maximum sensitivity of the matrix $\mathbf{A}$, and the row $\mathbf{a}_{i^\star}$ is the row of $\mathbf{A}$ with maximum sensitivity with respect to $\mathbf{A}$. Suppose the matrix $\mathbf{S}_\infty\mathbf{A}$ contains the row $\mathbf{a}_{i^\star}$. Then we have

$$\max_{i:\mathbf{c}_i\in\mathbf{S}_\infty\mathbf{A}}\boldsymbol{\sigma}_p^{\mathbf{S}_p\mathbf{A}}(\mathbf{c}_i) = \max_{\mathbf{x}\in\mathbb{R}^d,\mathbf{c}_k\in\mathbf{S}_\infty\mathbf{A}}\frac{|\mathbf{c}_k^\top\mathbf{x}|^p}{\|\mathbf{S}_p\mathbf{A}\mathbf{x}\|_p^p} = \Theta(1)\max_{\mathbf{x}\in\mathbb{R}^d,\mathbf{c}_k\in\mathbf{S}_\infty\mathbf{A}}\frac{|\mathbf{c}_k^\top\mathbf{x}|^p}{\|\mathbf{A}\mathbf{x}\|_p^p} = \Theta(1)\max_{\mathbf{x}\in\mathbb{R}^d}\frac{|\mathbf{a}_{i^\star}^\top\mathbf{x}|^p}{\|\mathbf{A}\mathbf{x}\|_p^p},$$
$$\tag{C.8}$$

---

[3]For $p=2$, the result of [23] gives a constant factor approximation to all leverage scores in $\mathbf{nnz}(\mathbf{A}) + d^\omega$ time.

where the first step is by definition of $\ell_p$ sensitivity of $\mathbf{S}_\infty \mathbf{A}$ with respect to $\mathbf{S}_p \mathbf{A}$, the second step is by Equation (C.6), and the third step by noting that the matrix $\mathbf{S}_\infty \mathbf{A}$ is a subset of the rows of $\mathbf{A}$ (which includes $\mathbf{a}_{i^\star}$). By definition of $\boldsymbol{\sigma}_p(\mathbf{a}_{i^\star})$ in Equation (C.7), we have

$$\max_{\mathbf{x}\in\mathbb{R}^d} \frac{|\mathbf{a}_{i^\star}^\top \mathbf{x}|^p}{\|\mathbf{A}\mathbf{x}\|_p^p} = \|\boldsymbol{\sigma}_p(\mathbf{A})\|_\infty. \tag{C.9}$$

Then, combining Equation (C.8) and Equation (C.9) gives the guarantee in this case. In the other case, suppose $\mathbf{a}_{i^\star}$ is not included in $\mathbf{S}_\infty \mathbf{A}$. Then we observe that the upper bound from the preceding inequalities still holds. For the lower bound, we observe that

$$\|\boldsymbol{\sigma}_p^{\mathbf{S}_p\mathbf{A}}(\mathbf{S}_\infty\mathbf{A})\|_\infty = \max_{\mathbf{x}\in\mathbb{R}^d, \mathbf{c}_j\in\mathbf{S}_\infty\mathbf{A}} \frac{|\mathbf{c}_j^\top \mathbf{x}|^p}{\|\mathbf{S}_p\mathbf{A}\mathbf{x}\|_p^p} \geq \max_{\mathbf{x}\in\mathbb{R}^d} \frac{\|\mathbf{S}_\infty\mathbf{A}\mathbf{x}\|_\infty^p}{\|\mathbf{S}_p\mathbf{A}\mathbf{x}\|_p^p} = \Theta(1) \max_{\mathbf{x}\in\mathbb{R}^d} \frac{\|\mathbf{S}_\infty\mathbf{A}\mathbf{x}\|_\infty^p}{\|\mathbf{A}\mathbf{x}\|_p^p}, \tag{C.10}$$

where the the second step is by choosing a specific vector in the numerator and the third step uses $\|\mathbf{S}_p\mathbf{A}\mathbf{x}\|_p^p = \Theta(\|\mathbf{A}\mathbf{x}\|_p^p)$. We further have,

$$\max_{\mathbf{x}\in\mathbb{R}^d} \frac{\|\mathbf{S}_\infty\mathbf{A}\mathbf{x}\|_\infty^p}{\|\mathbf{A}\mathbf{x}\|_p^p} \geq \frac{\|\mathbf{S}_\infty\mathbf{A}\mathbf{x}^\star\|_\infty^p}{\|\mathbf{A}\mathbf{x}^\star\|_p^p} \geq \frac{\|\mathbf{A}\mathbf{x}^\star\|_\infty^p}{d^{p/2}\|\mathbf{A}\mathbf{x}^\star\|_p^p} \geq \frac{|\mathbf{a}_{i^\star}^\top \mathbf{x}^\star|^p}{d^{p/2}\|\mathbf{A}\mathbf{x}^\star\|_p^p} = \frac{1}{d^{p/2}}\|\boldsymbol{\sigma}_p(\mathbf{A})\|_\infty, \tag{C.11}$$

where the first step is by choosing $\mathbf{x} = \mathbf{x}^\star$, the second step is by the guarantee of $\ell_\infty$ subspace embedding, and the final step is by definition of $\boldsymbol{\sigma}_p(\mathbf{a}_{i^\star})$. Combining Inequality (C.10) and Inequality (C.11) gives the claimed lower bound on $\|\boldsymbol{\sigma}_p^{\mathbf{S}_p\mathbf{A}}(\mathbf{S}_\infty\mathbf{A})\|_\infty$. The runtime follows from the cost of computing $\mathbf{S}_\infty\mathbf{A}$ from [35] and the cost of computing and size of $\mathbf{S}_p\mathbf{A}$ from [23, Figure 1] □

## D   Lower Bounds

**Theorem D.1** ($\ell_p$ **Regression Reduces to** $\ell_p$ **Sensitivities**)**.** *Suppose that we are given an algorithm $\mathcal{A}$, which for any matrix $\mathbf{A}' \in \mathbb{R}^{n'\times d'}$ and accuracy parameter $\varepsilon' \in (0,1)$, computes $(1\pm\varepsilon')\boldsymbol{\sigma}_p(\mathbf{A}')$ in time $\mathcal{T}(n', d', \mathbf{nnz}(\mathbf{A}'), \varepsilon')$. Then, there exists an algorithm that takes $\mathbf{A} \in \mathbb{R}^{n\times d}$ and $\mathbf{b} \in \mathbb{R}^n$ as inputs and computes $(1\pm\varepsilon)\min_{\mathbf{y}\in\mathbb{R}^d} \|\mathbf{A}\mathbf{y}-\mathbf{b}\|_p^p \pm \lambda^p\varepsilon$ in time $\mathcal{T}(n+1, d+1, \mathbf{nnz}(\mathbf{A})+\mathbf{nnz}(\mathbf{b})+1, \varepsilon)$ for any $\lambda > 0$.*

*Proof.* Given $\mathbf{A} \in \mathbb{R}^{n\times d}$ and $\mathbf{b} \in \mathbb{R}^d$, consider the matrix $\mathbf{A}' := \begin{bmatrix} \mathbf{A} & -\mathbf{b} \\ \mathbf{0}^\top & -\lambda \end{bmatrix} \in \mathbb{R}^{(n+1)\times d}$. Then the $n+1$-th $\ell_p$ sensitivity of $\mathbf{A}'$ is

$$\boldsymbol{\sigma}_p(\mathbf{a}'_{n+1}) = \max_{\mathbf{x}\in\mathbb{R}^d} \frac{|\langle\mathbf{e}_d, \mathbf{x}\rangle|^p}{\|\mathbf{A}'\mathbf{x}\|_p^p} = \max_{\mathbf{x}\in\mathbb{R}^d: x_d = \lambda^{-1}} \frac{1}{\|\mathbf{A}'\mathbf{x}\|_p^p} = \frac{1}{1 + \min_{\mathbf{y}\in\mathbb{R}^{d-1}}\{\|\mathbf{A}\mathbf{y} - \lambda^{-1}\mathbf{b}\|_p^p\}},$$

where the second step is by the scale-invariance of the definition of sensitivity with respect to the variable of optimization. By scale-invariance we see that for any $c > 0$, and $\min_{\mathbf{y}} \|\mathbf{A}_{:-i}\mathbf{y} - c\mathbf{A}_{:i}\|_p^p = c^p \min_{\mathbf{y}} \|\mathbf{A}_{:-i}\mathbf{y} - \mathbf{A}_{:i}\|_p^p$.

Therefore, note that rearranging gives $\min_{\mathbf{y}\in\mathbb{R}^{d-1}} \|\mathbf{A}\mathbf{y} - \mathbf{b}\|_p^p = \frac{\lambda^p}{\boldsymbol{\sigma}_p(\mathbf{a}'_{n+1})} - \lambda^p$, which is how we can estimate the cost of the regression. Then computing $\boldsymbol{\sigma}_p(\mathbf{a}'_{n+1})$ to a multiplicative accuracy of $\varepsilon'$ gives the value of $\min_{\mathbf{y}\in\mathbb{R}^{d-1}} \|\mathbf{A}\mathbf{y}-\mathbf{b}\|_p^p$ up to error $\varepsilon(\frac{\lambda^p}{\boldsymbol{\sigma}_p(\mathbf{a}'_{n+1})}) = \varepsilon(\lambda^p + \min_{\mathbf{y}\in\mathbb{R}^{d-1}} \|\mathbf{A}\mathbf{y}-\mathbf{b}\|_p^p)$, giving our final bounds.

□

Using the same technique as in Theorem D.1, we can similarly extend the reduction to a multiple regression task. In particular, we show that computing the values of a family of certain regularized leave-one-out regression problems for a matrix may be obtained by simply computing leverage scores of an associated matrix; therefore, this observation demonstrates that finding fast algorithms for sensitivities is as hard as multiple regression tasks. Specifically, for some matrix $\mathbf{A} \in \mathbb{R}^{n\times d}$, we denote $\mathbf{A}_{:-i} \in \mathbb{R}^{n\times d-1}$ as the submatrix of $\mathbf{A}$ with its $i$-th column, denoted as $\mathbf{A}_{:i}$, removed. We show that approximate sensitivity calculations can solve $\min_{\mathbf{y}\in\mathbb{R}^{d-1}} \|\mathbf{A}_{:-i}\mathbf{y} + \mathbf{A}_{:i}\|_2$ approximately for all $i$.

**Lemma D.2 (Regularized Leave-One-Out $\ell_p$ Multiregression Reduces to $\ell_p$ Sensitivities).** *Suppose that we are given an algorithm $\mathcal{A}$, which for any matrix $\mathbf{A}' \in \mathbb{R}^{n' \times d'}$ and accuracy parameter $\varepsilon' \in (0,1)$, computes $(1 \pm \varepsilon')\boldsymbol{\sigma}_p(\mathbf{A}')$ in time $\mathcal{T}(n', d', \mathbf{nnz}(\mathbf{A}'), \varepsilon')$. Given a matrix $\mathbf{A} \in \mathbb{R}^{n \times d}$ with $n \geq d$, let $OPT_i := \min_{\mathbf{y} \in \mathbb{R}^{d-1}} \|\mathbf{A}_{:-i}\mathbf{y} + \mathbf{A}_{:i}\|_p^p$ and $\mathbf{y}_i^\star := \arg\min_{\mathbf{y} \in \mathbb{R}^{d-1}} \|\mathbf{A}_{:-i}\mathbf{y} + \mathbf{A}_{:i}\|_p$ for all the $i \in [d]$. Then, there exists an algorithm that takes $\mathbf{A} \in \mathbb{R}^{n \times d}$ and computes $(1 \pm \varepsilon)OPT_i \pm \lambda^p(1 + \|\mathbf{y}_i^\star\|_p^p)$ in time $\mathcal{T}(n + d, d, \mathbf{nnz}(\mathbf{A}), \varepsilon)$ for any $\lambda > 0$.*

*Proof.* Given $\mathbf{A} \in \mathbb{R}^{n \times d}$, consider the matrix $\mathbf{A}' := \begin{bmatrix} \mathbf{A} \\ \lambda \mathbf{I} \end{bmatrix} \in \mathbb{R}^{(n+d) \times d}$ formed by vertically appending a scaled identity matrix to $\mathbf{A}$, with $\lambda > 0$. By the definition of $\ell_p$ sensitivities, we have the following.

$$\boldsymbol{\sigma}_p(\mathbf{a}'_{n+i}) = \max_{\mathbf{x} \in \mathbb{R}^d} \frac{|\mathbf{x}^\top \mathbf{a}'_{n+i}|^p}{\|\mathbf{A}'\mathbf{x}\|_p^p} = \max_{\mathbf{x} \in \mathbb{R}^d : x_i = \lambda^{-1}} \frac{1}{\|\mathbf{A}'\mathbf{x}\|_p^p} = \max_{\mathbf{x} \in \mathbb{R}^d : x_i = \lambda^{-1}} \frac{1}{\|\lambda^{-1}\mathbf{A}'_{:i} + \mathbf{A}'_{:-i}\mathbf{x}_{-i}\|_p^p}$$
$$= \frac{1}{\min_{\mathbf{y} \in \mathbb{R}^{d-1}} \{1 + \lambda^p\|\mathbf{y}\|_p^p + \|\mathbf{A}_{:-i}\mathbf{y} + \lambda^{-1}\mathbf{A}_{:i}\|_p^p\}}$$

Recall that $\mathbf{y}_i^\star := \arg\min_{\mathbf{y} \in \mathbb{R}^{d-1}} \|\mathbf{A}_{:-i}\mathbf{y} + \mathbf{A}_{:i}\|_p$, and since the power is a monotone transform, $OPT_i := \min_{\mathbf{y} \in \mathbb{R}^{d-1}} \|\mathbf{A}_{:-i}\mathbf{y} + \mathbf{A}_{:i}\|_p = \|\mathbf{A}_{:-i}\mathbf{y}_i^\star + \mathbf{A}_{:i}\|_p$. By scale-invariance we see that for any $c > 0$, $c\mathbf{y}_i^\star = \arg\min_{\mathbf{y} \in \mathbb{R}^{d-1}} \|\mathbf{A}_{:-i}\mathbf{y} + c\mathbf{A}_{:i}\|_p^p$ and $\min_{\mathbf{y}} \|\mathbf{A}_{:-i}\mathbf{y} + c\mathbf{A}_{:i}\|_p^p = c^p OPT_i$. Therefore, by plugging $\mathbf{y} = \lambda^{-1}\mathbf{y}_i^\star$, note that

$$\min_{\mathbf{y} \in \mathbb{R}^{d-1}} \left\{ 1 + \lambda^p\|\mathbf{y}\|_p^p + \|\mathbf{A}_{:-i}\mathbf{y} + \lambda^{-1}\mathbf{A}_{:i}\|_p^p \right\} \leq 1 + \|\mathbf{y}^\star\|_p^p + \lambda^{-p}OPT_i.$$

However, we also note that by non-negativity,

$$\min_{\mathbf{y} \in \mathbb{R}^{d-1}} \left\{ 1 + \lambda^p\|\mathbf{y}\|_p^p + \|\mathbf{A}_{:-i}\mathbf{y} + \lambda^{-1}\mathbf{A}_{:i}\|_p^p \right\} \geq \lambda^{-p}OPT_i.$$

Combining this all together gives the claimed approximation guarantee by noting that

$$OPT_i + \lambda^p\|\mathbf{y}^\star\|_p^p + \lambda^p \geq \frac{\lambda^p}{\boldsymbol{\sigma}_p(\mathbf{a}'_{n+i})} \geq OPT_i$$

Therefore, our algorithm is to simply return $\lambda^p / s_p(\mathbf{a}'_{n+i})$, where $s$ is the $\epsilon$-approximate estimate of the sensitivity and our bound follows. The runtime guarantees follow immediately from assumption that the runtime of calculating the $\ell_p$ sensitivities of $\mathbf{A}'$. $\qquad\square$

**Corollary D.3.** *Assuming that leave-one-out $\ell_p$ multi-regression with $\Omega(d)$ instances takes $\mathsf{poly}(d)\,\mathbf{nnz}(\mathbf{A}) + \mathsf{poly}(1/\varepsilon)$ time [44, 45], computing $\ell_p$ sensitivities of an $n \times d$ matrix $\mathbf{A}$ costs at least $\mathsf{poly}(d)\,\mathbf{nnz}(\mathbf{A}) + \mathsf{poly}(1/\varepsilon)$.*

# E  Additional Experiments

Here we include additional experiments on other similar datasets.

| $p$ | Total Sensitivity Upper Bound | Brute-Force | Approximation | Brute-Force Runtime (s) | Approximate Runtime (s) |
|---|---|---|---|---|---|
| 1 | 11 | 4.7 | 3.9 | 1940 | 303 |
| 1.5 | 11 | 8.3 | 10.3 | 1980 | 280 |
| 2.5 | 20 | 10.3 | 11.8 | 1970 | 326 |
| 3 | 36.4 | 6.9 | 7.3 | 1970 | 371 |

Table 2: Runtime comparison for computing total sensitivities for the `fires` dataset, which has matrix shape $(517, 11)$. We include the theoretical upper bound for the total sensitivities using lewis weights calculations.

| $p$ | Total Sensitivity Upper Bound | Brute-Force | Approximation | Brute-Force Runtime (s) | Approximate Runtime (s) |
|-----|------------------|-------------|---------------|------------------------|------------------------|
| 1   | 11   | 4.1  | 5.2  | 540 | 154 |
| 1.5 | 11   | 10.1 | 6.7  | 560 | 201 |
| 2.5 | 20   | 19.9 | 12.2 | 390 | 117 |
| 3   | 36.4 | 8.8  | 6.3  | 354 | 136 |

Table 3: Runtime comparison for computing total sensitivities for the `concrete` dataset, which has matrix shape $(101, 11)$. We include the theoretical upper bound for the total sensitivities using lewis weights calculations.

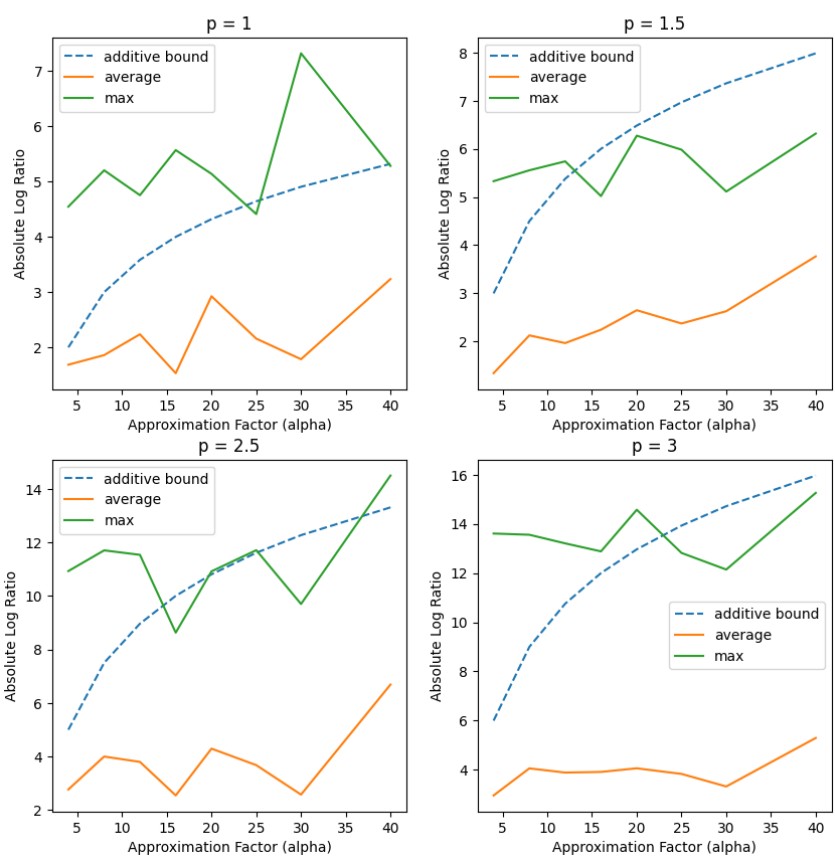

Figure 2: Average absolute log ratios for all $\ell_p$ sensitivity approximations for `fires`.

