# OpenReview forum: "Computing Approximate $\ell_p$ Sensitivities"
_NeurIPS.cc/2023/Conference — NeurIPS 2023 poster_

### Official Review · Reviewer_RJDW · 2023-07-04

**Soundness:** 3 good
**Presentation:** 3 good
**Contribution:** 3 good
**Rating:** 7
**Confidence:** 3

**Summary:**

The authors propose randomized algorithms to approximate $\ell_p$ sensitivity functions for $p \in [1,\infty)$, which extend leverage scores beyond the $\ell_2$ norm. The functions they consider are: 1. Estimating all sensitivities, 2. Estimating total sensitivity, and 3. Estimating maximum sensitivity. They provide different types of approximations for each task.

For task 1, they give an additive error, constant factor approximation. For task 2, they give a relative error, $(1+O(\gamma))$ approximation ($\gamma \in (0,1)$). And for task 3, they give a constant factor relative error approximation.

They demonstrate their algorithm initially for $\ell_1$ and then generalize it to all $\ell_p$ norms with $p > 1$. They also prove a hardness result by reducing $\ell_p$ regression to $\ell_p$ sensitivity estimation. Additionally, they implement their algorithm to estimate all sensitivities on 2 existing datasets and compare the average and maximum approximation ratios with the theoretical results.

The main techniques used in the algorithms involve hashing using Rademacher combinations, computing sensitivities with respect to subspace embeddings, splitting matrix rows based on leverage score intervals, and utilizing existing results for $\ell_\infty$ subspace embeddings.

**Strengths:**

The problem of sensitivity estimation is useful for regression problems and has received limited attention for general $p$. Approximating total sensitivity is particularly significant in obtaining the sample complexity of learning arbitrary functions.

The algorithms presented are an interesting combination of known results from sensitivity sampling framework and RNLA. Specifically, their result for approximating total sensitivity is interesting because the computational complexity does not depend polynomially on the number of rows of the matrix, which is usually very large.

Finally, the analysis of the proposed algorithms is largely clear, helping to understand the guarantees of their algorithms. Overall, this paper is a significant contribution to the field of sensitivity sampling and dimensionality reduction.

**Weaknesses:**


- Algorithm 4 does not appear to be significantly novel, except for integrating generalized sensitivity to an existing $\ell_\infty$ subspace embedding technique, but the task is coherent with the other tasks mentioned.
- While the motivation for total sensitivity is clear, the authors have not motivated the problem of estimating maximum sensitivity. However, this might be because I am not familiar with it.
- I noticed some typos, missing definitions.. In Algorithm 3, it is unclear where the vector of leverage scores $\tau(C)$ is being used. Additionally, $\omega$ is not defined, which I am assuming to be the matrix multiplication exponent.

**Questions:**

I would like to understand the comparison between approximate $\ell_p$ sensitivities and Lewis weights. From what I understand, it is known that Lewis weights cover sensitivities, at least for $p \in [1,2]$. The authors mentioned that Lewis weights are a crude approximation to sensitivities, although Lewis weights are used as a subroutine to obtain subspace embedding in their proposed algorithms. I am curious to see the benefits that an additive error approximation to sensitivities have over Lewis weight sampling.

**Limitations:**

It would be beneficial if the authors consider adding a section addressing the open problems and limitations.

---

> ### Author Rebuttal · Authors · 2023-08-08
>
> We are very grateful to the reviewer for their time, effort, and feedback. We are encouraged that they found the problem we study important and our algorithm and techniques novel. We are also very grateful for all the weaknesses and typos pointed out and questions raised, and we will clarify all these points in our manuscript.
>
> -----------------------
>
> ### “Usefulness of the maximum sensitivity”. ###
> The maximum sensitivity **captures the importance of the most important datapoint** and finds applications in, e.g., experiment design to detect the most important features and in reweighting matrices for low coherence [4]. Additionally, it **captures the maximum extent to which a datapoint can influence the objective function**, thus finding applications in differential privacy [1].
>
> ---------------------------
>
> ### “Where is $\tau(C)$ used in Algorithm 3?”. ###
> The leverage scores of $M$ (which is just a submatrix of $C$) are used critically in Lines 10-14 to split the rows into buckets. This step is actually quite subtle: The sensitivity of the $j^{th}$ row of $M$ with respect to $C$, i.e., $\sigma_1^{C}(M[j])$, is multiplicatively approximately the sensitivity of that row with respect to $SA$, which in turn is multiplicatively approximately the sensitivity of that row with respect to $A$. Further, the sensitivity of $M[j]$ with respect to $C$ can be sandwiched between appropriately scaled leverage scores of $C$. Therefore, we combine these two facts to obtain upper and lower bounds on the sensitivity of that row with respect to $A$. We formalize this notion in Lines 685-699 of the Supplementary material but will clarify the role of $\tau(C)$ better in the main text.
>
> --------------------------
>
> ### “Using additive approximate sensitivities instead of Lewis weights”. ###
> For the case $p > 2$, by using Theorem $1.5$ of [3], we have that $$\text{the sample complexity with approximate sensitivities} = O\left(\alpha^{2p} \cdot \mathfrak{S}_p^{2-2/p}(A)\right),$$ as opposed to $$\text{the sample complexity with true sensitivities} = O\left(\mathfrak{S}_p^{2-2/p}(A)\right),$$ and $$\text{the sample complexity with Lewis weights} = O\left(d^{p/2}\right).$$ Assume that $p>2$ is large and the total sensitivity $\mathfrak{S}_p(A)$ is small (say, $\mathfrak{S}_p(A) = d$). Further assume, as a toy example, $n = d^{10}$ and $\alpha = n^{\frac{1}{10 p}} = d^{\frac{1}{p}}$. Then, our approximate sensitivities give a sample complexity of $O(d^4)$, true sensitivities give a sample complexity of $O(d^2)$, and Lewis weights sampling gives $O(d^{p/2})$. Thus, our approximate sensitivities preserve the regression approximation guarantee, while **increasing the total sample complexity by only a small $\text{poly}(d)$ compared to that with true sensitivities (while still being much smaller than that with Lewis weights) and incurring a much smaller cost.** A detailed discussion of the connection of Lewis weights to well-conditioned bases can be found in Section $3.3$ of [4].
>
> -----------------
>
> ### Conclusion ###
>
> We once again thank the reviewer for their time, effort, and very thoughtful questions. We are happy to provide any further clarification required.
>
> ------------------
>
> ### References ###
>
> [1] “The Algorithmic Foundations of Differential Privacy”, Cynthia Dwork and Aaron Roth, Foundations and Trends in Theoretical Computer Science 2014
>
> [2] “Uniform Sampling for Matrix Approximation”, Michael B. Cohen, Yin Tat Lee, Cameron Musco, Christopher Musco, Richard Peng, Aaron Sidford, ITCS 2015
>
> [3] “Sharper Bounds for $\ell_p$ Sensitivity Sampling”, David P. Woodruff and Taisuke Yasuda, ICML 2023
>
> [4] "Dimensionality Reduction for Tukey Regression", Kenneth L. Clarkson, Ruosong Wang, and David P. Woodruff, ICML 2019, https://arxiv.org/abs/1905.05376

---

> > ### Comment · Reviewer_RJDW · 2023-08-21
> >
> > Thanks for such a well organized rebuttal. Your comments have clarified my questions.

---

### Official Review · Reviewer_Vy4U · 2023-07-06

**Soundness:** 3 good
**Presentation:** 2 fair
**Contribution:** 3 good
**Rating:** 7
**Confidence:** 4

**Summary:**

The paper gives algorithms to compute, approximately, the individual sensitivity scores, total sensitivity and maximum sensitivity, for $\ell_p $ norms. The approximation for the individual sensitivity scores is additive while a relative error approximation is obtained for both total and maximum sensitivity. Calculating exact sensitivity scores often being a computationally expensive problem, the authors provide faster algorithms that use techniques of multiplication with Rademacher vectors and hashing. The authors also empirically validate their results on real datasets.

**Strengths:**

Strengths of the paper
1) Methods of constructing coresets for various ML problems rely heavily on good approximation to sensitivity scores. As such the paper is important and will be of interest to the community.
2) Empirical Results for various values of $p$ which are not usually seen in literature.

**Weaknesses:**

To me the main weakness appears to be with the writing and clarity of the paper (may be because of space constraints) and also to an extent comparison and discussion with some related works. Here I list out some of the questions/ suggestions that I have:
1) The authors mention Lewis weights as crude over approximation to sensitivity scores for $\ell_p$. However, another way to sample rows for $\ell_p$ subspace embeddings is using the row norms of a well-conditioned basis. The authors have not discussed much from the area. It would be interesting to compare if for some well-conditioned basis, the row norms correspond to some approximation of sensitivity scores.
2) In all the algorithms the authors rely on constructing SA which is an $\ell_1$ subspace embedding. There is good amount of work on $\ell_1$ subspace embeddings using 1-stable Cauchy random variables, exponential random variables etc. It has not been compared with. It would be useful to give a table that compares these methods in terms of time, approximation factors and no. of rows required etc. also how is SA calculated in this work and what is the time taken for it? Please clarify.
3) Also, algorithm 2 requires computation of leverage scores of A. Are they calculated exactly in which case the time required will be $O(nd^2)$ and if they are also approximated, how does the approximation factor figure in your guarantees?
4) The implications of the lower bound are not clarified. Please elaborate.
5) It would be useful to give some motivation as to why maximum of the sensitivity scores is important.

Overall, the paper uses some known techniques in Randomized numerical linear algebra literature to calculate the approximations to $\ell_p$ sensitivities. However improved writing, better clarification of exact contributions and comparison with existing literature will strengthen it.

**Questions:**

Please see weaknesses section.

**Limitations:**

Please see weaknesses section.

---

> ### Author Rebuttal · Authors · 2023-08-08
>
> We are extremely thankful to the reviewer for raising a very thorough set of questions and comments and will incorporate the clarifications for each of them in our manuscript. We are encouraged by the reviewer’s assessment of our work as important and interesting for the community. We address below the reviewer's questions and concerns.
>
> ------------------------
>
> ### How is $SA$ computed? ###
>  We use the results of Cohen-Peng [2] to construct subspace embeddings. These incur a cost of only $O(\text{nnz}(A))$, which is the best possible cost (since that is the cost of merely reading the input data matrix), and so we do not compare with other methods. However, we acknowledge that there exist many methods for this task (such as the ones noted by the reviewer) and will incorporate them in our "Related Work" section to provide a more complete picture of the landscape on subspace embeddings.
>
> ----------------------
>
> ### Calculation of leverage scores. ###
> As was shown in [1], leverage scores can be computed up to a constant factor approximation in time $O(\text{nnz}(A) + d^{\omega})$ (we are ignoring polylogarithmic factors here). This is the accuracy to which we compute leverage scores in our submission; indeed, since the error accumulates multiplicatively across different steps of the algorithm, a high-accuracy algorithm does not serve any benefit over this constant-accuracy one.
>
> -------------------
>
> ### “Implication of lower bound”. ###
> Our hardness results imply that in general one cannot compute all sensitivities as quickly as leverage scores unless there is a major breakthrough in regression. We will make this point clearer in our write-up.
>
> ---------------------
>
> ### “Usefulness of the maximum sensitivity”. ###
> As detailed in our top-level response, computing the maximum sensitivity is useful in applications ranging from experiment design to differential privacy [3] to more generic pre-processing tasks such as reducing the coherence of the matrix, which was studied for $p=2$ in [4].
>
> -------------------------------
>
> ### “Row norms of a well-conditioned basis” ###
> It is indeed true that the Lewis basis is a well-conditioned basis, but as was shown in [5], there are many advantages that sensitivity sampling offers over Lewis weight sampling, such as a much lower sample complexity for $p>2$ or when the total sensitivity is small. Additionally, while other well-conditioned bases exist and can be computed in $\text{poly}(d)$ time, they result in worse $\text{poly}(d)$ factor distortion, as was shown in, e.g., [4].
>
> -------------------------
>
> ### Conclusion ###
>
> We again thank the reviewer for their time and effort in bringing up these questions; we will incorporate these answers in our manuscript.
>
> Please let us know if any questions remain, and if we answered all the questions, we’d like to respectfully request that the reviewer re-consider their score of our submission. We are happy to answer any further questions.
>
> -------------------------
>
> ### References. ###
>
> [1] “Uniform Sampling for Matrix Approximation”, Michael B. Cohen, Yin Tat Lee, Cameron Musco, Christopher Musco, Richard Peng, Aaron Sidford, ITCS 2015
>
> [2] "$\ell_p$ Row Sampling by Lewis Weights", Michael B. Cohen and Richard Peng, STOC 2015
>
> [3] “The Algorithmic Foundations of Differential Privacy”, Cynthia Dwork and Aaron Roth, Foundations and Trends in Theoretical Computer Science 2014
>
> [4] "Dimensionality Reduction for Tukey Regression", Kenneth L. Clarkson, Ruosong Wang, and David P. Woodruff, ICML 2019
>
> [5] "Sharper Bounds for $\ell_p$ Sensitivity Sampling", David P. Woodruff and Taisuke Yasuda, ICML 2023

---

> ### Comment · Reviewer_Vy4U · 2023-08-17
> **Replying to Rebuttal**
>
> Thanks for the response. It clears my doubts and I have raised my score

---

### Official Review · Reviewer_T77r · 2023-07-07

**Soundness:** 3 good
**Presentation:** 3 good
**Contribution:** 3 good
**Rating:** 7
**Confidence:** 2

**Summary:**

This paper proposes a randomized algorithm for efficiently approximating the $\ell_p$ sensitivities, with a constant approximation parameter guaranteed.

**Strengths:**

This paper presents several novel randomized algorithms for approximating $\ell_p$ sensitivities and related statistics, based on two key ideas:

(1) Using subspace embeddings to efficiently approximate $\|Ax\|_p$ that avoids computing $Ax$ on the whole dataset

(2) Randomly hashing the dataset into small subsets and computing the sensitivities for each subset separately

Given the above, this work provides efficient approximations of the sensitivities of all data samples, the total sensitivity, and the maximum one. The resulting algorithms are good contributions to the problem of estimating sensitivities that fill in the blank of computing $\ell_p$ sensitivities efficiently.

The theoretical guarantees appear to be solid and correct, which is further verified by several experiments.




**Weaknesses:**



**Questions:**

I am a bit confused about the definition of $\boldsymbol{S}$ in line 191, where the size of $\boldsymbol{S}$ is $r$ by $d$, should it be $r$ by $n$? Also, in lines 189 and 193 it is said $\boldsymbol{S}$ to be 'diagonal' while this seems to contradict its definition, please let me know if I miss anything.

---

> ### Author Rebuttal · Authors · 2023-08-08
>
> We are very grateful to the reviewer for their time spent reviewing our submission and are encouraged by their positive assessment of the motivation of our work, our theory, and our experiments.
>
> ---------------
>
> In response to the reviewer’s question: Yes, thank you for pointing out the typo in Line 191; the size of $S$ should be $r \times n$. We apologize for the confusion about $S$: one can think of $S$ as being a sparse diagonal matrix, and in the resulting matrix $SA$, delete all the zero rows. We hope this clarifies the structure of $S$ and will incorporate these fixes into our manuscript.
>
> ---------------
>
> We again thank the reviewer for their time and effort. Please let us know if there are any further questions that we could help clarify!

---

### Official Review · Reviewer_Y5nz · 2023-07-09

**Soundness:** 3 good
**Presentation:** 3 good
**Contribution:** 2 fair
**Rating:** 5
**Confidence:** 3

**Summary:**

Given a matrix $A \in \mathbb{R}^{n \times d}$ and $p \in (0, \infty)$, the $l_p$ sensitivity of a vector $a \in \mathbb{R}^d$ with respect to $A$ is defined as $\sigma_p(a) := \max_x |a^\top x|^p / |Ax|_p^p$. It is known that by sampling each row $a_i$ of $A$ with probability proportional to (an upper bound on) $\sigma_p(a_i)$, we can obtain a coreset of $A$ with respect to the $l_p$ loss. Therefore, estimating $\sigma_p(a_i)$ quickly is desirable. This work presents three fast approximation algorithms for $l_p$ sensitivities. The main focus is on the case where $p = 1$, so I will state the results specifically for that case:

- An algorithm that provides an estimate $\tilde{\sigma}$ such that $\sigma_1(a_i) \leq \tilde{\sigma} \leq \sigma_1(a_i) + \alpha/n \mathfrak{S}_1(A)$, where $\mathfrak{S}_1(A)$ is the sum of $l_1$ sensitivities. The algorithm's running time is $O(n/\alpha (nnz(A) + d^\omega) + n)$, where $\omega$ is the matrix multiplication exponent.
- An algorithm that provides an estimate $\tilde{\sigma}$ such that $\mathfrak{S}_1(A) \leq \tilde{\sigma} \leq (1 + \gamma)\mathfrak{S}_1(A)$. The running time of this algorithm is $O(\sqrt{d} (nnz(A) + d^\omega))$, significantly faster than the naive bound of $O(n (nnz(A) + d^\omega))$.
- An algorithm that provides an estimate $\tilde{\sigma}$ such that $\max_i \sigma_1(a_i) \leq \tilde{\sigma} \leq \sqrt{d} \max_i \sigma_1(a_i)$. The running time for this algorithm is $O(d(nnz(A) + d^\omega))$, also faster than the naive bound.

The authors showed that $l_p$ regression reduces to $l_p$ sensitivity calculation, implying that designing a fast algorithm for $l_p$ sensitivity requires a fast algorithm for $l_p$ regression. Experimental results are provided to validate the theoretical bounds.

**Strengths:**

The algorithm effectively utilizes $l_p$ subspace embedding to enhance its speed. Additionally, the recursive approach employed for computing total sensitivity is intriguing.

**Weaknesses:**

- There are reservations about the usefulness of Theorem 1.2. It is possible to estimate sensitivities in $O(n (nnz(A) + d^\omega))$ time, as indicated in Fact 2.4. This means that, to achieve a significant speedup, $\alpha$ needs to be a function of $n$, which then results in a substantial additive error.
- The usefulness of Theorem 1.3 is unclear. Although the sum provides a bound on the coreset's size, it is not evident when one would solely desire the bound without the coreset itself.
- Similarly, the usefulness of Theorem 1.4 is not clear.

**Questions:**

- This is just a comment, but "additive error of $O(\alpha^p)$" should be "additive error of $\alpha^p/n \cdot \mathfrak{S}_p(A)$".
- What is the meaning of the "$\rho$-factor subspace embedding" in Algorithm 2? Should it be "$\rho$-approximate subspace embedding"?


**Limitations:**

The limitations are not clearly mentioned.

---

> ### Author Rebuttal · Authors · 2023-08-08
>
> We are deeply grateful to the reviewer for their careful reading of our manuscript and for raising very thoughtful questions, which we respond to below.
>
> -----------------
>
> ### ”Usefulness of Theorem 1.2”: Runtime ###
> **We would like to clarify that our actual runtime is significantly better than what we (mistakenly) first stated in the main submission.** Our sub-optimal runtimes existed only in our theorems, and the algorithms and analyses originally presented indeed give the correct runtimes (e.g., cf. Appendix of the Supplementary file, Lines 551 - 575). Specifically, our correct main result (for the case $p=1$, cf. Lines 73-74 of the Supplementary Material) is that we can approximate all $\ell_1$ sensitivities at a cost of $$ O\left(\text{nnz}(A) + \frac{n}{\alpha} \cdot d^\omega\right), $$ with an approximation guarantee $\sigma_1(a_i) \leq \widetilde{\sigma}_1(a_i)\leq \frac{\alpha}{n} \mathfrak{S}_1(A)$ for all $i\in [n]$. As the reviewer noted in their review, naively computing all sensitivities would cost $O(n \cdot (\text{nnz}(A) + d^\omega))$. Thus, **our obtained runtime greatly improves over the trivial and (to our best knowledge) only known result**. Our proof sketch for this runtime uses the construction of $\ell_1$ subspace embeddings [4] and reduction to solving a $d\times d$ linear program (and applying [5] to solve it). Please see the top-level response for the details.
>
> -----------------
>
> ### "Usefulness of Theorem 1.2": Sample Complexity ###
> For the case $p > 2$, by using Theorem $1.5$ of [1], we have that $$\text{the sample complexity with approximate sensitivities} = O\left(\alpha^{2p} \cdot \mathfrak{S}_p^{2-2/p}(A)\right),$$ as opposed to $$\text{the sample complexity with true sensitivities} = O\left(\mathfrak{S}_p^{2-2/p}(A)\right),$$ and $$\text{the sample complexity with Lewis weights} = O\left(d^{p/2}\right).$$ Assume that $p>2$ is large and the total sensitivity $\mathfrak{S}_p(A)$ is small (say, $\mathfrak{S}_p(A) = d$). Further assume, as a toy example, $n = d^{10}$ and $\alpha = n^{\frac{1}{10 p}} = d^{\frac{1}{p}}$. Then, our approximate sensitivities give a sample complexity of $O(d^4)$, true sensitivities give a sample complexity of $O(d^2)$, and Lewis weights sampling gives $O(d^{p/2})$. Thus, our approximate sensitivities preserve the regression approximation guarantee, while **increasing the total sample complexity by only a small $\text{poly}(d)$ compared to that with true sensitivities (while still being much smaller than that with Lewis weights) and incurring a much smaller cost.**
>
> -----------------
>
> ### ”Usefulness of Theorem $1.3$”. ###
>  Please refer to our top-level response for the full answer; Briefly, one potential application of Theorem $1.3$ is to run the corresponding algorithm to determine the total sensitivity and, only if this quantity is substantially smaller than expected, we may use sensitivity sampling; this is based on the result by Woodruff and Yasuda [1] that in cases with low total sensitivity, sensitivity sampling significantly outperforms Lewis weights sampling.
>
> -----------------
>
> ### ”Usefulness of Theorem 1.4”. ###
>  The maximum sensitivity **captures the importance of the most important datapoint** and finds applications in, e.g., experiment design to detect the most important features and in reweighting matrices for low coherence, which was studied for $p=2$ in [6]. Additionally, it captures the maximum extent to which a datapoint can influence the objective function, thus finding applications in differential privacy [3].
>
> -----------------
>
> ### Clarifying notation. ###
>  Yes, $\rho$-factor subspace embedding means a $\rho$-approximate subspace embedding, i.e., a matrix $S$ such that for all $x$, we have $\|SAx\|_p^p \in (1\pm \rho) \|Ax\|_p^p$. We will include this in the notation section and propagate the change wherever applicable.
>
> -----------------
>
> ### Conclusion. ###
> We again thank the reviewer for their time and effort in bringing up very pertinent questions; we will incorporate these answers in our manuscript.
>
> Please let us know if any questions remain, and if we answered all the questions, we’d like to respectfully request that the reviewer re-consider their score of our submission. We are happy to answer any further questions.
>
> -----------------
>
> ### References ###
>
> [1] “Sharper Bounds for $\ell_p$ Sensitivity Sampling”, David P. Woodruff and Taisuke Yasuda, ICML 2023
>
> [2] “Uniform Sampling for Matrix Approximation”, Michael B. Cohen, Yin Tat Lee, Cameron Musco, Christopher Musco, Richard Peng, Aaron Sidford, ITCS 2015
>
> [3] “The Algorithmic Foundations of Differential Privacy”, Cynthia Dwork and Aaron Roth, Foundations and Trends in Theoretical Computer Science 2014
>
> [4] "$\ell_p$ Row Sampling by Lewis Weights", Michael B. Cohen and Richard Peng, STOC 2015
>
> [5] "Solving Linear Programs in the Current Matrix Multiplication Time", Michael B. Cohen, Yin Tat Lee, and Zhao Song, STOC 2019
>
> [6] "Dimensionality Reduction for Tukey Regression", Kenneth L. Clarkson, Ruosong Wang, and David P. Woodruff, ICML 2019

---

> > ### Comment · Reviewer_Y5nz · 2023-08-11
> >
> > The response addresses some of my concerns, and I'll increase the score.

---

### Author Rebuttal · Authors · 2023-08-08

# Top-Level Response #

We thank all the reviewers for their time, effort, and suggestions. Here, we restate some of our key contributions and answer common questions. We also reply to each reviewer individually.

---

## Motivating the problem ##
### Why sensitivities? ###
A common preprocessing step for $\ell_p$ regression ($\min_{x} \|Ax\|_p^p$) involves constructing an $\ell_p$ subspace embedding of matrix $A$. This is usually done via a sampling matrix $S$ that ensures $\|S A x\|_p^p = (1\pm \epsilon) \|Ax\|_p^p$ for all vectors $x$. This reduces our problem's data dimension from the number of rows of $A$ to that of $SA$.

A recent result [1] shows that **a sampling matrix $S$ built using the $\ell_p$ sensitivities of $A$ is much smaller than one built using its $\ell_p$ Lewis weights in many important cases**, e.g., when $p > 2$, or when the total sensitivity is small. Consequently, sensitivity sampling offers significant advantages over Lewis weight sampling, which also extends to many ML applications beyond regression [1, Section 1.1].

However, until our work, **there were no known efficient algorithms to approximate $\ell_p$ sensitivities for general $p>0$.** Our paper initiates a systematic study of algorithms for this task.

### Why total sensitivity? (asked by R1). ###
As alluded to above, when the total sensitivity is small, the sample complexity of sensitivity sampling is much lower than that of Lewis weight sampling [1]. Hence, a quick approximation to the total sensitivity can be used as **a fast test for whether or not to proceed with sensitivity sampling (involving the costly task of calculating all sensitivities).**

Further, for $\ell_2$ sensitivities, the total sensitivity is the rank, which may be used in rank estimation subroutines [6]. The total $\ell_p$ sensitivity is analogous to $\ell_p$ rank, which may be similarly useful.

### Why maximum sensitivity? (asked by R1, R3, and R4). ###
 The maximum sensitivity **captures the importance of the most important datapoint** and is used in, e.g., experiment design and in reweighting matrices for low coherence [7]. Additionally, it **captures the maximum extent to which a datapoint can influence the objective function** and is used in differential privacy [5].

---

## Clarification on our results. ##
We first clarify that some statements of our results in the main submission were correct but sub-optimally stated — we have corrected these in our full version (cf. Lines 510 - 519, Lines 72 - 73, Lines 83 - 84, and Line 89 of the Supplementary file). **These sub-optimal statements existed only in our theorems, and the algorithms and analyses originally presented indeed give the correct cost (e.g., cf. Lines 551 - 575 of the Supplementary file).**

### Computing all sensitivities. ###
Our correct main result for $p=1$ (cf. Lines 73-74 of the Supplementary Material) is that we can approximate all $\ell_1$ sensitivities (with additive error $\frac{\alpha}{n}\mathfrak{S}_1(A)$) at a cost of $ O(nnz(A) + (n/\alpha) \cdot d^\omega). $ As **R1** noted, the naive total cost is $O(n (nnz(A) + d^\omega))$. Thus, **our obtained result greatly improves over the naive and (to our best knowledge) only known result**.

### Proof sketch. ###
 As a reminder, the proof sketch for this cost is: Constructing an $\ell_1$ subspace embedding $SA$ costs $O(nnz(A))$ [3]  (asked by **R3**); plus, computing $n/\alpha$ sensitivities with respect to the $d \times d$ matrix $SA$ costs $(n/\alpha) \cdot d^\omega$ (by reducing one sensitivity computation w.r.t. $SA$ to solving a $d\times d$ linear program and using the current fastest LP solver [4]).

### Sampling with approximate sensitivities (asked by R1 and R4). ###
For $p > 2$, Theorem $1.5$ of [1] implies $$\text{the sample complexity with approximate sensitivities} = O\left(\alpha^{2p} \cdot \mathfrak{S}_p^{2-2/p}(A)\right),$$ as opposed to $$\text{the sample complexity with true sensitivities} = O\left(\mathfrak{S}_p^{2-2/p}(A)\right),$$ and $$\text{the sample complexity with Lewis weights} = O\left(d^{p/2}\right).$$ Assume a large $p>2$ and small total sensitivity, say, $\mathfrak{S}_p(A) = d$. Further assume, as a toy example, $n = d^{10}$ and $\alpha = n^{\frac{1}{10 p}} = d^{\frac{1}{p}}$. Then, sample complexity with approximate sensitivities is $O(d^4)$, with true sensitivities is $O(d^2)$, and with Lewis weights is $O(d^{p/2})$. Thus, our approximate sensitivities preserve the regression approximation guarantee, while **increasing the total sample complexity by only a small $\text{poly}(d)$ compared to that with true sensitivities (while still being much smaller than that with Lewis weights) and incurring a much smaller cost.**

### Computing the total and maximum sensitivities. ###
For the total and maximum $\ell_1$ sensitivities, our runtimes are, respectively, $ O(nnz(A) + d^{\omega + 1/2})$ and $O(nnz(A) + d^{\omega+1}),$ with **no polynomial dependence on $n$** and significantly better than the naive cost.

---
## References ##
[1] “Sharper Bounds for $\ell_p$ Sensitivity Sampling”, David P. Woodruff and Taisuke Yasuda, ICML 2023

[2] “Uniform Sampling for Matrix Approximation”, Michael B. Cohen, Yin Tat Lee, Cameron Musco, Christopher Musco, Richard Peng, Aaron Sidford, ITCS 2015

[3] "$\ell_p$ Row Sampling by Lewis Weights", Michael B. Cohen and Richard Peng, STOC 2015

[4] "Solving Linear Programs in the Current Matrix Multiplication Time", Michael B. Cohen, Yin Tat Lee, and Zhao Song, STOC 2019

[5] “The Algorithmic Foundations of Differential Privacy”, Cynthia Dwork and Aaron Roth, Foundations and Trends in Theoretical Computer Science 2014

[6] “Rank Estimation For (Approximately) Low-Rank Matrices”, Niloofar Bayat, Cody Morrin, Yuheng Wang, and Vishal Misra, ACM SIGMETRICS Performance Evaluation Review 2022

[7] "Dimensionality Reduction for Tukey Regression", Kenneth L. Clarkson, Ruosong Wang, and David P. Woodruff, ICML 2019

---

### Decision · Program_Chairs · 2023-09-21

**Decision:**

Accept (poster)

**Comment:**

This paper attacks the problem of computing l_p sensitivities using tools from approximation algorithms and randomized algorithms. This is related to other linear algebra problems which have attracted a lot of interest like estimating leverage scores and lewis weights. After the author-reviewer discussion, some important points of confusion/typos regarding the main results in the original draft were clarified, and a positive consensus was formed of the overall technical contribution. Based on this, I recommend acceptance of this work. Please make the promised edits in the next version of the paper.